# ZigZag: Universal Sampling-free Uncertainty Estimation Through Two-Step Inference

**Nikita Durasov**                                                    *nikita.durasov@epfl.ch*
*Computer Vision Laboratory, EPFL*

**Nik Dorndorf**                                                      *nik.dorndorf@rwth-aachen.de*
*RWTH Aachen*

**Hieu Le**                                                          *minh.le@epfl.ch*
*Computer Vision Laboratory, EPFL*

**Pascal Fua**                                                       *pascal.fua@epfl.ch*
*Computer Vision Laboratory, EPFL*

**Reviewed on OpenReview:** *https://openreview.net/forum?id=QSvb6jBXML*

## Abstract

Whereas the ability of deep networks to produce useful predictions on many kinds of data has been amply demonstrated, estimating the reliability of these predictions remains challenging. Sampling approaches such as MC-Dropout and Deep Ensembles have emerged as the most popular ones for this purpose. Unfortunately, they require many forward passes at inference time, which slows them down. Sampling-free approaches can be faster but often suffer from other drawbacks, such as lower reliability of uncertainty estimates, difficulty of use, and limited applicability to different types of tasks and data.

In this work, we introduce a sampling-free approach that is generic and easy to deploy, while producing reliable uncertainty estimates on par with state-of-the-art methods at a significantly lower computational cost. It is predicated on training the network to produce the same output with and without additional information about it. At inference time, when no prior information is given, we use the network's own prediction as the additional information. We then take the distance between the predictions with and without prior information as our uncertainty measure.

We demonstrate our approach on several classification and regression tasks. We show that it delivers results on par with those of ensembles but at a much lower computational cost.

## 1 Introduction

Though the ability of modern neural networks to generate accurate predictions is now clear, assessing the trustworthiness of these predictions remains an open problem. This can be addressed by estimating the potential inaccuracy of the predictions, which is then taken as an uncertainty measure. *MC-Dropout* (Gal & Ghahramani, 2016) and *Deep Ensembles* (Lakshminarayanan et al., 2017) are the most widely methods used to this end. MC-Dropout involves randomly zeroing out network weights and assessing the effect, whereas ensembles involves training multiple networks, starting from different initial conditions. They are simple to deploy and universal. Unfortunately, they induce substantial computational and memory overheads, which makes them unsuitable for many real-world applications.

An alternative is to use sampling-free methods that estimate uncertainty in one single forward pass of a single neural network, thereby avoiding computational overheads (Amersfoort et al., 2020; Malinin & Gales, 2018; Tagasovska & Lopez-Paz, 2018; Postels et al., 2019). However, deploying them may require significant

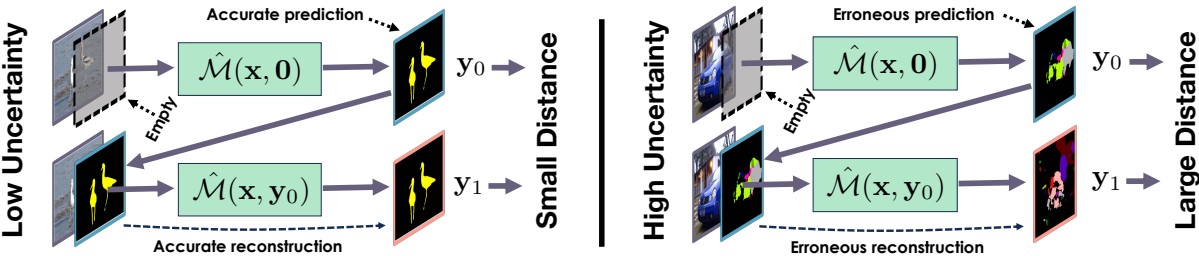

Figure 1: **ZigZaging.** At inference time, we make two forward passes. First, we use $[\mathbf{x}, \mathbf{0}]$ as input to produce a prediction $\mathbf{y}_0$. Second, we feed $[\mathbf{x}, \mathbf{y}_0]$ to the network and generate $\mathbf{y}_1$. We take $\|\mathbf{y}_0 - \mathbf{y}_1\|$ to be our uncertainty estimate. In essence, the second pass performs a reconstruction in much the same way an auto-encoder does and a high reconstruction error correlates with uncertainty.

modifications to the network's architecture (Postels et al., 2019), substantial changes to the training procedures (Malinin & Gales, 2018), limiting their application to very specific tasks (Amersfoort et al., 2020; Malinin & Gales, 2018; Mukhoti et al., 2021a), or lessen the quality of the uncertainty estimate (Postels et al., 2022; Ashukha et al., 2020). As a result, they have not gained as much traction as MC-Dropout and ensembles.

To remedy this, we introduce *ZigZag*, a sampling-free approach that is generic and easy to deploy, while producing reliable uncertainty estimates on par with sampling-based methods, but at a significantly lower computational cost. It only requires two forward passes through a single network. The first one simply predicts the output from the input data. The second one takes the initial prediction as an additional input to make a second prediction. The consistency between these two predictions is indicative of uncertainty. This is because we train the network to achieve accurate predictions in the first pass and to precisely reconstruct these predictions in the second pass, assuming that the first one is correct as shown in Fig.1 (Left). Conversely, if the initial prediction is erroneous, the subsequent reconstruction is likely to fail as shown in Fig.1 (Right). We will argue that this is analogous to using the reconstruction error of regular autodecoders to tell in-distribution samples from out-of-distribution ones (Japkowicz et al., 1995; Alain & Bengio, 2014; Zhou, 2022).

More specifically, given a network $\mathcal{M}$, we modify its first layer to accept a second argument, yielding the modified architecture $\hat{\mathcal{M}}$. We then train $\hat{\mathcal{M}}$ so that, for all training pairs $(\mathbf{x}, \mathbf{y})$, we have $\mathbf{y} \approx \hat{\mathcal{M}}(\mathbf{x}, \mathbf{0}) \approx \hat{\mathcal{M}}(\mathbf{x}, \mathbf{y})$, where $\mathbf{0}$ is a vector of zeros. At inference time, we first compute $\mathbf{y}_0 = \hat{\mathcal{M}}(\mathbf{x}, \mathbf{0})$ and then $\mathbf{y}_1 = \hat{\mathcal{M}}(\mathbf{x}, \mathbf{y}_0)$. We refer to this as ZigZagging, as depicted by Fig. 1. Finally, we take the distance between the two predictions $\|\mathbf{y}_0 - \mathbf{y}_1\|$ as our error estimate. This exploits the fact that, if $\mathbf{y}_0$ is accurate, that is, $\mathbf{y}_0 \approx \mathbf{y}$, $\mathbf{y}_1 = \hat{\mathcal{M}}(\mathbf{x}, \mathbf{y}_0)$ is likely to be close to $\mathbf{y}$ as well because that's what the network has been trained to do. Thus $\|\mathbf{y}_0 - \mathbf{y}_1\|$ will be small. A contrario, if $\mathbf{y}_0$ is *wrong* and very different from $\mathbf{y}$, feeding the pair $(\mathbf{x}, \mathbf{y}_0)$ to the network amounts to giving it an input that is out-of-distribution with respect to the data it has been trained to handle. Thus, the result is likely to be random and the distance between $\mathbf{y}_0$ and $\mathbf{y}_1$ large.

Our approach is fast because it only requires performing two forward passes using one single network and delivers uncertainty results comparable to those of ensembles, which are much more costly but often seen as the method that delivers the best uncertainty estimates on a wide range of classification and regression problems. Furthermore, it is very easy to use in conjunction with almost any network architecture with only very minor changes. Hence, our method is also task-agnostic. We demonstrate its effectiveness across a wide range of classification and regression tasks, extending to practical applications such as lift-drag regression for airfoil samples and predicting the drag coefficient of a 3D car.

## 2 Related work

Uncertainty Estimation (UE) aims to accurately evaluate the reliability of a model's predictions. Among all the methods that can be used to do this, MC-Dropout (Gal & Ghahramani, 2016) and Deep Ensembles (Lak-

shminarayanan et al., 2017) have emerged as two of the most popular ones, with Bayesian Networks (Mackay, 1995) being a third alternative. These methods are sampling-based and require several predictions at inference time, which slows them down. There is recent work on overcoming this and we discuss both kinds of approaches below.

**Sampling-based Approaches.** MC-Dropout involves randomly zeroing out network weights and assessing the effect, whereas ensembles involve training multiple networks, starting from different initial conditions. The extensive survey of Ashukha et al. (2020) concludes that Deep Ensembles tend to produce the most decorrelated models, which results in highly diversified predictions and the most reliable uncertainty estimates. Unfortunately, Deep Ensembles also entail the highest computation costs due to the need to train multiple networks and to run up to dozens of forward passes at inference time. MC-Dropout tends to be less reliable and also involves making several inferences at inference time. There have been recent attempts at increasing the reliability of MC-Dropout (Durasov et al., 2021a; Wen et al., 2020) but they do not address the fact that multiple inferences are required to estimate the uncertainty. Alternative sampling-based methods such as Mi et al. (2022) rely on noise injections or input augmentations during inference in order to produce uncertainty from the variance of generated predictions. Bayesian Networks (Blundell et al., 2015; Graves, 2011; Hernández-Lobato & Adams, 2015; Kingma et al., 2015) also require several forward passes to compute uncertainty and rarely outperform Deep Ensembles (Ashukha et al., 2020). In short, for sampling-based methods, computation time scales linearly with the number of samples and can be prohibitively expensive for performance-critical applications.

**Sampling-free Approaches.** When a rapid response is needed, for example for robotic control (Loquercio et al., 2020) or low-latency applications (Gal, 2016), there is no time to perform many forward passes during inference. Consequently, there has been much interest for sampling-free approaches that require *constant* time for inference. For example, (Amersfoort et al., 2020) describes a clustering-like procedure used to estimate uncertainty for classification and semantic segmentation purposes. In (Malinin & Gales, 2018; Sensoy et al., 2018; Amini et al., 2020; Malinin et al., 2020), uncertainty is estimated from Dirichlet and Normal-Wishart distributions whose parameters are predicted by the network. Unfortunately, it is not obvious how to extend sampling-free methods designed for specific tasks (Amersfoort et al., 2020; Mukhoti et al., 2021a; Hornauer & Belagiannis, 2022) to more generic applications. Furthermore, deploying them often requires significantly changing the network architecture and the training procedures (Liu et al., 2020; Shekhovtsov & Flach, 2019; Wannenwetsch & Roth, 2020), along with increased memory consumption (Wang et al., 2016; Shekhovtsov & Flach, 2019; Gast & Roth, 2018), worse uncertainty quality (Tagasovska & Lopez-Paz, 2018; Mukhoti et al., 2021b) or slower inference (Postels et al., 2019), which limits their appeal.

Regression is handled in (Postels et al., 2019; Gast & Roth, 2018) using an uncertainty estimation method that relies on uncertainty propagation from one layer to another. During the forward pass, not only activations but also their variances are estimated in each layer. Thus, the variance of the final predictions can be estimated in one pass but at the cost of a two-fold memory consumption and we will show that our approach performs better. Tagasovska & Lopez-Paz (2018) use both quantile regression (Furno & Vistocco, 2018) and *orthonormal certificates* to detect out-of-distribution samples during inference. Though being computationally efficient, this approach also can yield poor uncertainty estimates and miscalibrated predictions. SNGP (Liu et al., 2020) has been used to estimate uncertainty in a deep learning context, but this requires adding a Random Features (Rahimi & Recht, 2007) and Spectral Normalization (Miyato et al., 2018) layer to every convolutional layer, which entails significant modifications of both training dynamic and network architectures. Similarly, the approaches described by Wang et al. (2016); Shekhovtsov & Flach (2019) require replacing all of the convolution layers with modified versions and doubling the number of weights, which significantly impacts memory consumption. Other types of methods such as (Zhang et al., 2019) work only with specific uncertainty types, either aleatoric or epistemic (Der Kiureghian & Ditlevsen, 2009; Kendall & Gal, 2017). Furthermore, in some cases, they can yield significantly worse uncertainty calibration than sampling-based approaches (Postels et al., 2022). We provide more specific comparisons in the experiment section.

**Reconstruction Error in Autoencoders.** It has long been known that, for any sample fed as input to an autoencoder, the reconstruction error can be used to estimate the likelihood of this sample being

within the model's training distribution (Japkowicz et al., 1995), as depicted by Fig.2. Given a denoising or contractive autoencoder $\mathcal{R}$ and a sample $\mathbf{x}$ the reconstruction error $|\mathcal{R}(\mathbf{x}) - \mathbf{x}|$ is directly connected to the log-probability of the data distribution $p_{data}(\mathbf{x})$, as initially shown by Bengio et al. (2013); Alain & Bengio (2014). This finding was later extended to a wider class of autoencoders (Kamyshanska & Memisevic, 2013), and eventually to regular autoencoders trained using a stochastic optimization setup (Solinas et al., 2020). This has been successfully used for out-of-distribution detection (Zhou, 2022; Sabokrou et al., 2016) task. Our approach is in the same spirit, except for the fact we replace the reconstruction error $|\mathcal{R}(\mathbf{x}) - \mathbf{x}|$ by the distance between the two ZigZag predictions.

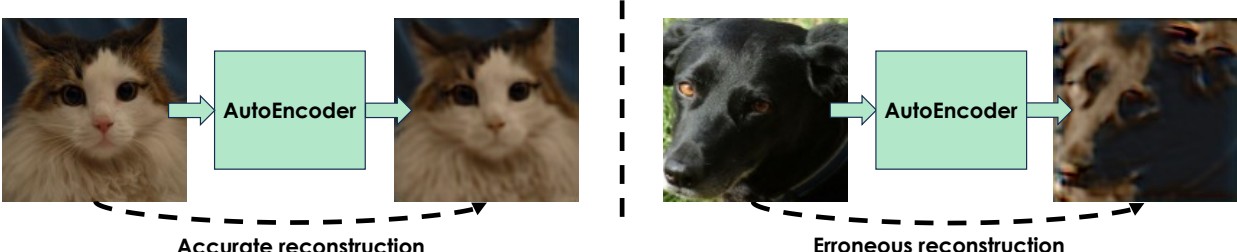

Figure 2: **Autoencoder Reconstruction Error** An autoencoder trained exclusively on cat images yields accurate reconstructions on other cat images (left) and inaccurate ones on dog images (right). Thus, the distance between an image and its reconstruction can be used to estimate whether that image is likely to be a cat image or not.

## 3 Method

*ZigZag* is an approach to sampling-free uncertainty estimation that delivers classification and regression results on par with state-of-the-art sampling-based methods such as ensembles and MC-Dropout while being far less computationally demanding. It relies on the dual-inference scheme depicted by Fig. 1. Given a network $\hat{\mathcal{M}}$ and a sample $\mathbf{x}$, we first use $\hat{\mathcal{M}}$ to make a first prediction $\mathbf{y}_0$ *without* any prior information. We then make a second prediction $\mathbf{y}_1$ using $\mathbf{y}_0$ as a prior. We train the network so that, if $\mathbf{y}_0$ is correct, then $\mathbf{y}_1$ should be similar to $\mathbf{y}_0$, whereas it should be different if $\mathbf{y}_0$ is inaccurate. In much the same way, an auto-encoder prediction is accurate for in-distribution samples and inaccurate for out-of-distribution ones.

In other words, the second pass performs a reconstruction of the second input argument. As in an auto-encoder, we expect the reconstruction error to be low for in-distribution data and high for out-of-distribution data, thereby providing an estimation of uncertainty. More specifically, when we provide the network $\hat{\mathcal{M}}(\mathbf{x}, \mathbf{0})$ with input $(\mathbf{x}, \mathbf{y})$, if the label $\mathbf{y}$ is close to the correct answer, then the difference between $\hat{\mathcal{M}}(\mathbf{x}, \mathbf{0})$ and $\hat{\mathcal{M}}(\mathbf{x}, \mathbf{y})$ is small because the network is trained to behave in this manner and the input $(\mathbf{x}, \mathbf{y})$ represents an in-distribution data point. However, if $\mathbf{y}$ is not close to the correct answer, then $(\mathbf{x}, \mathbf{y})$ represents something that the network has never encountered during training: a first argument $\mathbf{x}$ and a second argument $\mathbf{y}$ that is neither $\mathbf{0}$ nor the ground-truth. Essentially, this is an out-of-distribution sample for which the network, like most networks, can be expected to produce an unpredictable output. Hence, there is no reason for $\hat{\mathcal{M}}(\mathbf{x}, \mathbf{0})$ and $\hat{\mathcal{M}}(\mathbf{x}, \mathbf{y})$ to be similar. We leverage this property to quantify the model's uncertainty. In other words, there are two scenarios when reconstruction fails: 1) when $(\mathbf{x}, \mathbf{y})$ is OOD because $\mathbf{x}$ is OOD, addressing *epistemic* uncertainty and OOD samples, 2) when $(\mathbf{x}, \mathbf{y})$ is OOD because $\mathbf{y}$ is OOD / errornous. In this case, the reconstruction issue is due to $\mathbf{y}$, our uncertainty measure is high, we cover *aleatoric* uncertainty connected to predicted target.

### 3.1 Modifying the Original Architecture

Let $\mathcal{M}$ be a network that takes as input a vector $\mathbf{x}$ and returns a prediction vector $\mathcal{M}(\mathbf{x})$ that should be close to target $\mathbf{y}$. We modify the first layer of $\mathcal{M}$ to create a new architecture $\hat{\mathcal{M}}$ that takes as input both $\mathbf{x}$ and a vector of the same dimension as $\mathbf{y}$ so that we can compute both $\hat{\mathcal{M}}(\mathbf{x}, \mathbf{0})$ and $\hat{\mathcal{M}}(\mathbf{x}, \mathbf{y})$, where $\mathbf{0}$ is a vector of zeros of the same dimension as $\mathbf{y}$. We take $\hat{\mathcal{M}}(\mathbf{x}, \mathbf{0})$ to be the prediction without prior information

and $\hat{\mathcal{M}}(\mathbf{x}, \mathbf{y})$ one with prior information. In practice, $\mathcal{M}$ can be any sufficiently powerful deep architecture, such as VGG (Simonyan et al., 2014), ResNet (He et al., 2016) or a Transformer (Dosovitskiy et al., 2020). In all these cases, modifying the first network layer is a simple task.

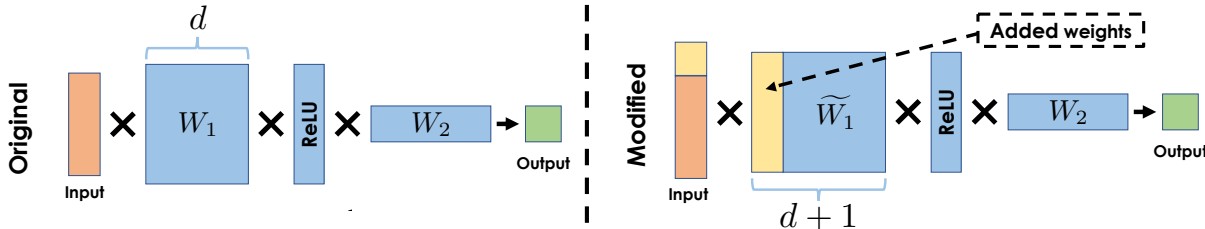

Figure 3: **Architecture Modification.** Given a model with weights $W_1 \in \mathbb{R}^{d \times h}, W_2 \in \mathbb{R}^{h \times 1}$, we modify its first layer $W_1$ to accept two inputs instead of only one. The modified model consists of $\widetilde{W_1} \in \mathbb{R}^{(d+1) \times h}$ and $W_2 \in \mathbb{R}^{h \times 1}$ and can process the concatenation of the original input $\mathbf{x}$ and additional value $\mathbf{y_0}$.

For simplicity, let us first consider the case where $\mathcal{M}$ is a simple network with one hidden layer of dimension $h$. It takes $\mathbf{x} \in \mathbb{R}^d$ as input and outputs a scalar $y \in \mathbb{R}$. To handle a second argument $\mathbf{y}$, the input dimension of the first trainable layer must become $d + 1$ to allow the concatenation of the original input vector $\mathbf{x}$ and the additional value $\mathbf{y}$. Similarly, we can add additional channels to convolutional layers to work with RGB images, for example by adding a fourth channel that represents $\mathbf{y}$. As before, we only need to modify the first convolutional layer of the network so that it can process 4-dimensional inputs.

## 3.2 Training

At the heart of our approach is the training of $\hat{\mathcal{M}}$ to yield comparable outputs, whether or not the target $\mathbf{y}$ is provided as input. In practice, we want $\hat{\mathcal{M}}(\mathbf{x}, \mathbf{0}) \approx \hat{\mathcal{M}}(\mathbf{x}, \mathbf{y}) \approx \mathbf{y}$ for all training pairs $(\mathbf{x}, \mathbf{y})$. To this end, given a training pair$(\mathbf{x}, \mathbf{y})$, we consider the loss

$$\mathcal{L}(\hat{\mathcal{M}}(\mathbf{x}, \mathbf{0}), \mathbf{y}) + \mathcal{L}(\hat{\mathcal{M}}(\mathbf{x}, \mathbf{y}), \mathbf{y}) \ , \tag{1}$$

where $\mathcal{L}$ is a domain-dependent loss term whose minimization ensures that the prediction made by $\mathcal{M}$ is close to the target $\mathbf{y}$. By minimizing this loss for all samples, we guarantee that our model can make accurate predictions when provided either with $\mathbf{0}$, that is, no prior information, or with accurate prior information in the form of the full answer $\mathbf{y}$.

## 3.3 Inference

At inference time, we compute

$$\hat{\mathbf{y}} = \mathbf{y_0} = \hat{\mathcal{M}}(\mathbf{x}, \mathbf{0}) \ , \tag{2}$$

$$\mathbf{y_1} = \hat{\mathcal{M}}(\mathbf{x}, \mathbf{y_0}) = \hat{\mathcal{M}}(\mathbf{x}, \hat{\mathcal{M}}(\mathbf{x}, \mathbf{0})) \ ,$$

$$\hat{\mathbf{u}} = \|\mathbf{y_0} - \mathbf{y_1}\| \ , \tag{3}$$

where $\hat{\mathbf{y}}$ is our final prediction and $\hat{\mathbf{u}}$ our uncertainty estimate. $\mathbf{y_0}$ is estimated without prior information, whereas $\mathbf{y_1}$ is computed by using $\mathbf{y_0}$ as the prior information. If $\mathbf{y_0}$ is accurate, providing it as prior information should not disturb the inference because $\hat{\mathcal{M}}$ has been trained to return the correct answer when given the correct answer as prior information. This should result in $\mathbf{y_1}$ being similar to $\mathbf{y_0}$ and a small $\hat{\mathbf{u}}$. Consequently a large $\hat{\mathbf{u}}$ is an indication that supplying $\mathbf{y_0}$ as the prior information has disrupted the inference mechanism and that $\mathbf{y_0}$ is probably inaccurate.

This can also be understood in terms of the well-known tendency of networks to return arbitrary answers for samples that are out-of-distribution (OOD) with respect to the data they were trained on (Zhang et al., 2021; Lakshminarayanan et al., 2017; Nguyen et al., 2015). The training scheme introduced above is such

that the in-distribution training pairs are of the form $(\mathbf{x}, \mathbf{z})$, where $\mathbf{z}$ is either $\mathbf{0}$ or the ground-truth $\mathbf{y}$. When $\mathbf{y}_0$ is inaccurate and we use it to compute $\mathbf{y}_1 = \hat{\mathcal{M}}(\mathbf{x}, \mathbf{y}_0)$, we are essentially feeding an OOD sample to the network and the result $\mathbf{y}_1$ can be expected to be random, and therefore very different in general of $\mathbf{y}_0$. Our experiments on validation data bear this out, as shown in Fig. 4.

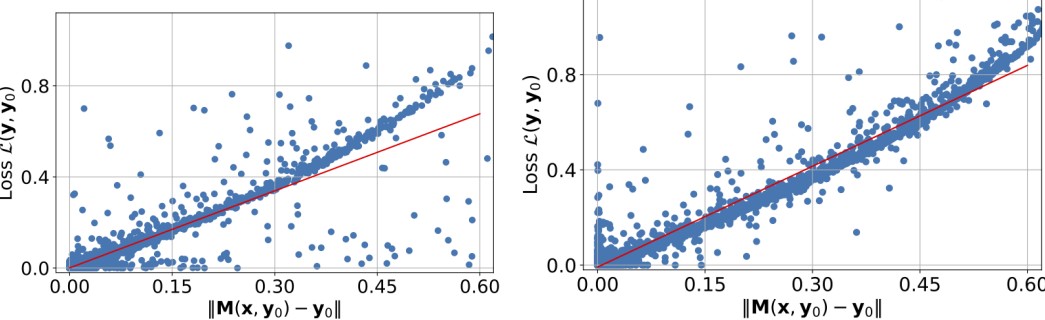

Figure 4: **True vs Estimated Error.** We use MNIST (left) and CIFAR (right) validation data to plot the true prediction errors as measured by the loss being minimized against our uncertainty estimates $\|\hat{\mathcal{M}}(\mathbf{x}, \mathbf{0}) - \hat{\mathcal{M}}(\mathbf{x}, \hat{\mathcal{M}}(\mathbf{x}, \mathbf{0}))\|$ for individual samples. In both cases, the correlation is strong and Pearson's correlation coefficient is above 90%. The red line represents a linear fit to the data.

## 4 Experiments

We first introduce our metrics and baselines. We then use simple synthetic data to illustrate the behavior of *ZigZag*. Next, we turn to image datasets often used to test uncertainty-estimation algorithms. Finally, we present real-world applications. Implementation details about the baselines, metrics, training setups, and hyper-parameters could be found in the appendix.

### 4.1 Metrics and Baselines

We now introduce the evaluation metrics we use to quantitatively compare our methods against several baselines.

**Accuracy Metric.** For classification tasks, we use the standard classification accuracy, i.e. the percentage of correctly classified samples in the test set. For regression tasks, we use the standard *Mean Absolute Error* (MAE) metric.

**Uncertainty Metrics.** As in (Postels et al., 2022), we use *Relative Area Under the Lift Curve* (rAULC) to quantify the quality of calibration of uncertainty measures both for classification and regression tasks. Unlike other metrics (Brier et al., 1950; Guo et al., 2017), it is suitable for sampling-free approaches to estimate uncertainty.

Another way to estimate how good uncertainty estimates are is to use them to detect out-of-distribution samples under the assumption that the network is more uncertain about those than about samples from the distribution used to train it. As in (Malinin & Gales, 2018; Durasov et al., 2021a), given both in- and out-of-distribution (OOD) samples, we classify high-uncertainty ones as OOD and rely on standard classification metrics, ROC and PR AUCs, to quantify the classification performance.

**Time, Memory, and Simplicity.** The *Time* and *Size* metrics measure how much time and memory it takes to train the network(s) to estimate uncertainty, compared to a single one that does not estimate it. The *Simplicity* metric assesses how easy it is to modify a given architecture to obtain uncertainty estimates. We denote it as simple (✔) if it requires changing less than 10% of layers of the original model and the training procedures do not need to be substantially modified. We also report *Inference Time* that represents

how much time the model takes to compute uncertainties relative to single model inference on Tesla V100 and without considering parallelization for sampling-based approaches.

**Baselines.** We compare against recent sample-based approaches—MC-Dropout (Gal & Ghahramani, 2016) (*MC-D*), Deep Ensembles (Lakshminarayanan et al., 2017) (*DeepE*), BatchEnsemble (Wen et al., 2020) (*BatchE*) and Masksembles (Durasov et al., 2021a) (*MaskE*) —and sample-free ones—-Single Model (Kendall & Gal, 2017) (*Single*)—-predicted variance for regression and entropy of the predictive distribution for classification, Orthonormal Certificates (Tagasovska & Lopez-Paz, 2019) (*OC*), SNGP (Liu et al., 2020) (*SNGP*), EDL (Sensoy et al., 2018) (*EDL*), Variance Propagation (Postels et al., 2019) (*VarProp*). For all four sampling-based approaches, we use five samples to estimate the uncertainty at inference time. This number of samples has been shown to perform well for many tasks (Durasov et al., 2021a; Wen et al., 2020; Malinin & Gales, 2018). All of the training and implementation details are provided in the appendix. The evaluation was performed using three random seeds and averaged. For standard deviation results, please refer to Sec. A.3.

### 4.2 Simple Synthetic Data.

We use such data to illustrate how *ZigZag* behaves both for classification and regression.

**Classification Task.** Let us consider the red and blue 2D points shown in Fig. 5. We use them to train an MLP with 6 fully-connected layers and LeakyReLU activations to classify other points in the plane as belonging either to the red or the blue class. The background color in each of the subplots depicts the uncertainty estimated by Single-Model, MC-Dropout, Deep Ensembles, and *ZigZag*. The first two only exhibit uncertainty along a narrow band between the two distributions. This is highly questionable once one is far away from the data points in the lower left and upper right corners of the range. Both ensembles and *ZigZag* deliver more plausible high uncertainties once far from the training points, but *ZigZag* does it at a lower computational cost.

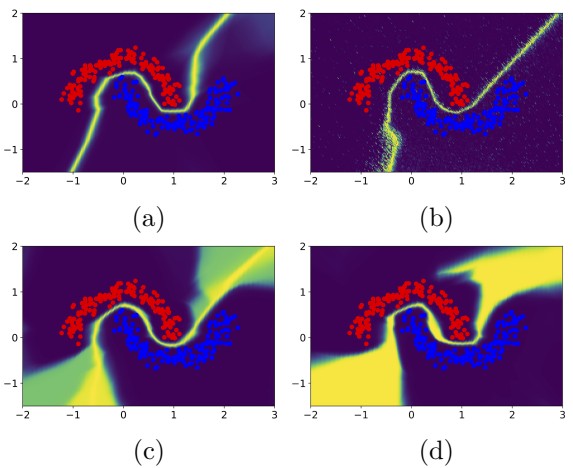

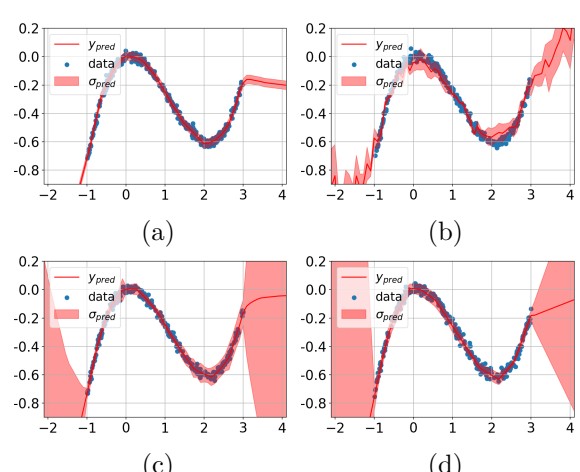

Figure 5: **Uncertainty Estimation for Classification.** The task is to classify data points drawn in the range $x \in [-2, 3]$, $y \in [-2, 2]$ as being red or blue given the red and blue training samples from two interleaving half circles with added Gaussian noise. The background color depicts the classification uncertainty assigned by different techniques to individual grid points. Violet is low and yellow is high. **(a)** Single model, **(b)** MC-Dropout, **(c)** Deep Ensembles, **(d)** *ZigZag*.

Figure 6: **Uncertainty Estimation for Regression.** The task is to regress $y$-axis values for $x$-axis data points drawn in the range $x \in [-1, 3]$ from third power polynomial with added Gaussian noise. Red colored area depicts the uncertainty assigned by different models to individual points on the $x$-axis grid. **(a)** Single model, **(b)** MC-Dropout, **(c)** Deep Ensembles, **(d)** *ZigZag*.

**Regression Task.** Let us now consider the simple regression problem depicted by Fig. 6: We draw values $x$ in the range $[-1, 3]$, compute values $y = f(x) + \sigma$ where $f$ is the third order polynomial and $\sigma$ is Gaussian noise, and use these pairs to train a regression network. The shaded areas depict the uncertainty estimated

by Single-Model, MC-Dropout, Deep Ensembles, and *ZigZag*. For the points outside of the training range, the last two correctly predict very large uncertainties, unlike the first two. But again, *ZigZag* does it at a lower computational cost than Deep Ensembles.

| | MC-D | DeepE | BatchE | MaskE | Single | EDL | OC | SNGP | VarProp | *ZigZag* | |
|---|---|---|---|---|---|---|---|---|---|---|---|
| Accuracy (↑) | 0.981 | **0.990** | 0.989 | 0.989 | 0.980 | 0.975 | 0.980 | 0.984 | 0.986 | 0.982 | |
| rAULC (↑) | 0.932 | 0.958 | 0.941 | 0.929 | 0.712 | 0.955 | 0.851 | 0.813 | 0.731 | **0.961** | |
| Size | 1x | 5x | 1.2x | 1x | 1x | 1x | 1.3x | 1x | 1x | 1x | MNIST |
| Inf. Time | 5x | 5x | 5x | 5x | 1x | 1x | 1.4x | 1.7x | 1.2x | 2x | |
| Time | 1.3x | 5x | 1.4x | 1.3x | 1x | 1x | 1.1x | 1.1x | 1.x | 1x | |
| ROC-AUC (↑) | 0.953 | **0.984** | 0.965 | 0.963 | 0.773 | 0.947 | 0.934 | 0.951 | 0.812 | 0.982 | |
| PR-AUC (↑) | 0.962 | 0.979 | 0.965 | 0.966 | 0.844 | 0.923 | 0.923 | 0.942 | 0.861 | **0.981** | |
| Accuracy (↑) | 0.909 | **0.929** | 0.911 | 0.901 | 0.8901 | 0.912 | 0.892 | 0.905 | 0.895 | 0.928 | |
| rAULC (↑) | 0.889 | **0.911** | 0.884 | 0.889 | 0.884 | 0.596 | 0.583 | 0.742 | 0.715 | 0.897 | |
| Size | 1x | 5x | 1.2x | 1x | 1x | 1x | 1x | 1x | 1x | 1x | CIFAR |
| Inf. Time | 5x | 5x | 5x | 5x | 1x | 1x | 1.1x | 1.1x | 1.2x | 2x | |
| Time | 1.2x | 5x | 1.4x | 1.3x | 1x | 1x | 1.3x | 1x | 1.2x | 1.2x | |
| ROC-AUC (↑) | 0.854 | **0.915** | 0.877 | 0.900 | 0.825 | 0.864 | 0.851 | 0.900 | 0.831 | 0.901 | |
| PR-AUC (↑) | 0.918 | **0.949** | 0.919 | 0.931 | 0.875 | 0.903 | 0.821 | 0.891 | 0.861 | 0.933 | |
| Accuracy (↑) | 0.74 | **0.77** | 0.73 | 0.74 | 0.75 | 0.74 | 0.75 | 0.74 | 0.73 | 0.75 | |
| rAULC (↑) | 0.78 | **0.84** | 0.78 | 0.79 | 0.80 | 0.76 | 0.71 | 0.8 | 0.69 | 0.82 | |
| Size | 1x | 5x | 1.1x | 1.1x | 1x | 1x | 1x | 1x | 1x | 1x | ImageNet |
| Inf. Time | 5x | 5x | 5x | 5x | 1x | 1x | 1.1x | 1x | 1x | 2x | |
| Time | 1x | 5x | 1.1x | 1x | 1x | 1x | 1x | 1x | 1.1x | 1.3x | |
| ROC-AUC (↑) | 0.52 | **0.56** | 0.53 | 0.52 | 0.51 | 0.52 | 0.52 | 0.53 | 0.50 | 0.54 | |
| PR-AUC (↑) | 0.16 | **0.19** | 0.16 | 0.14 | 0.15 | 0.14 | 0.17 | 0.13 | 0.12 | 0.17 | |
| Simple | ✓ | ✓ | ✗ | ✗ | - | ✓ | ✓ | ✗ | ✗ | ✓ | |

Table 1: **Classification results on MNIST (top), CIFAR (middle), and ImageNet (bottom).** The best result in each category is in **bold** and the second best is in **bold**. Most correspond to *ZigZag* and *DeepE*. Hence, they perform similarly but *ZigZag* requires far less computation and memory.

## 4.3 Classification Tasks.

We now compare *ZigZag* against the baselines on the widely used benchmark datasets MNIST vs FMNIST, CIFAR vs SVHN, and ImageNet vs ImageNet-O. The images are very different across datasets and exhibit distinct statistics. For each dataset pair, we use the first to train the network and to evaluate how well calibrated the methods are the second to perform out-of-domain detection experiments. We report the results in Tab 1. In all three cases, Deep Ensembles and *ZigZag* perform similarly and outperform the other approaches. However, *ZigZag* does *not* incur the 5-fold increase in memory and time requirements that Deep Ensembles does. Even though the other sampling-free approaches tend to yields worse calibration than sampling-based ones (Postels et al., 2022), *ZigZag* does not.

**MNIST vs FashionMNIST.** We train the networks on MNIST (LeCun et al., 1998) and compute the accuracy and calibration metrics. We then use the uncertainty measure they produce to classify images from the test sets of MNIST and FashionMNIST (Xiao et al., 2017) as being within the MNIST distribution or not to compute the OOD metrics introduced above. We use a standard architecture with four convolution and pooling layers, followed by fully connected layers with ReLU activations.

**CIFAR vs SVHN.** We ran a similar experiment with the CIFAR10 (Krizhevsky et al., 2014) and SVHN (Netzer et al., 2011) datasets. This is challenging for OOD detection because many of the CIFAR $32 \times 32$ images are noisy and therefore hardly distinguishable from each other, which makes the class labels unreliable. As the training set is relatively small, very large models tend to overfit the training data. We therefore use the *Deep Layer Aggregation* (DLA) (Yu et al., 2018) network for our experiments that tends to outperform standard architectures such as ResNet (He et al., 2016) or VGG (Simonyan & Zisserman,

| | MC-D | DeepE | BatchE | MaskE | Single | SNGP | OC | VarProp | *ZigZag* | |
|---|---|---|---|---|---|---|---|---|---|---|
| MAE (↓) | 4.655 | **4.472** | 4.699 | 4.786 | 4.724 | 4.819 | 4.724 | 4.682 | 4.630 | UTKFACE |
| rAULC (↑) | 0.034 | **0.047** | 0.043 | 0.033 | 0.026 | 0.031 | 0.025 | 0.029 | 0.045 | |
| Size | 1x | 5x | 1x | 1x | 1x | 1x | 1x | 1x | 1x | |
| Inf. Time | 5x | 5x | 5x | 5x | 1x | 1x | 1x | 1x | 2x | |
| Time | 1.1x | 5x | 1.3x | 1.2x | 1x | 1.7x | 1.1x | 1x | 1x | |
| ROC-AUC (↑) | 0.688 | 0.755 | 0.732 | 0.653 | 0.564 | 0.694 | 0.658 | 0.662 | **0.773** | |
| PR-AUC (↑) | 0.884 | 0.939 | 0.846 | 0.830 | 0.762 | 0.890 | 0.843 | 0.851 | **0.959** | |
| MAE (↓) | 3.376 | **3.03** | 3.03 | 3.26 | 3.18 | 3.16 | 3.20 | 3.25 | 3.10 | AIRFOILS |
| rAULC (↑) | 0.065 | 0.062 | 0.062 | 0.034 | 0.008 | 0.013 | 0.01 | 0.015 | **0.068** | |
| Size | 1x | 5x | 1x | 1x | 1x | 1.05x | 1x | 1x | 1x | |
| Inf. Time | 5x | 5x | 5x | 5x | 1x | 1.3x | 1.1x | 1.4x | 2x | |
| Time | 1.2x | 5x | 1.3x | 1.3x | 1x | 1.7x | 1.1x | 1.1x | 1.2x | |
| ROC-AUC (↑) | 0.897 | 0.972 | 0.971 | 0.923 | 0.690 | 0.894 | 0.874 | 0.78 | **0.992** | |
| PR-AUC (↑) | 0.744 | 0.955 | 0.942 | 0.793 | 0.681 | 0.767 | 0.748 | 0.76 | **0.987** | |
| MAE (↓) | 0.129 | **0.101** | 0.115 | 0.115 | 0.121 | 0.115 | 0.120 | 0.119 | 0.112 | CARS |
| rAULC (↑) | 0.06 | **0.10** | 0.08 | 0.07 | 0.03 | 0.06 | 0.04 | 0.03 | 0.07 | |
| Size | 1x | 5x | 1.05x | 1.05x | 1x | 1x | 1x | 1x | 1x | |
| Inf. Time | 5x | 5x | 5x | 5x | 1x | 1.3x | 1.1x | 1.2x | 2x | |
| Time | 1.1x | 5x | 1.2x | 1.3x | 1x | 1.2x | 1x | 1.1x | 1.2x | |
| ROC-AUC (↑) | 0.851 | 0.954 | 0.921 | 0.926 | 0.755 | 0.872 | 0.831 | 0.816 | **0.956** | |
| PR-AUC (↑) | 0.734 | 0.941 | 0.867 | 0.832 | 0.534 | 0.751 | 0.723 | 0.567 | **0.974** | |
| Simple | ✓ | ✓ | ✗ | ✗ | - | ✓ | ✗ | ✗ | ✓ | |

Table 2: **Regression results on Age Prediction (top), Airfoils (middle) and Cars (bottom).** As in Table 1 the best two results in each category are shown in bold and correspond to *ZigZag* and *DeepE*, except in terms of computation time and memory requirements where *ZigZag* does much better.

2015) and trained it as recommended in the original paper. We sample our out-of-distribution data from the SVHN dataset that comprises images belonging to classes that are not in CIFAR10, such as road sign digits.

**ImageNet vs ImageNet-O**   We experimented with the ImageNet dataset (Russakovsky et al., 2015) and its counterpart, ImageNet-O (Hendrycks et al., 2021). The latter is an extension of the original ImageNet dataset, which is designed to evaluate the robustness and generalization capabilities of machine learning models by providing a challenging set of images that are difficult to classify correctly. As in the CIFAR vs. SVHN scenario, this sets up a challenge for Out-of-Distribution (OOD) detection. We use a standard ResNet50 architecture and the training setup from (He et al., 2016).

**Influence of the Number of Samples.**   The five-fold increase in computation time that the sampling-based methods incur is a direct consequence of ours using 5 samples. *ZigZag* performs two inferences, which is equivalent to using 2 samples. Thus, in Fig 7, we plot OOD classification performance as a function of the number of samples used. Given only 2 and 3 samples, all sampling-based methods do worse than *ZigZag*. With 4 or 5, Deep Ensembles is the only one that matches or beats us. However, it then needs at least quadruple the computational budget and memory.

### 4.4   Regression Tasks.

We report similar results for three very different regression tasks in Tab. 2. As for classification tasks, we use data samples significantly different from training ones for OOD evaluation. Then, after generating uncertainty for ID and OOD samples, we evaluate standard AUC metrics as for classification problems. The overall behavior is similar to what we observed for classification. *ZigZag* performs on par with Deep Ensembles and better than the others, while being much less computationally demanding than Deep Ensembles.

**Age Prediction.**   First, we consider image-based age prediction from face images. To this end, we use UTKFace (Zhang et al., 2017), a large-scale dataset containing tens of thousands of face images annotated with associated age information. We use a network with a large ResNet-152 backbone and five linear layers

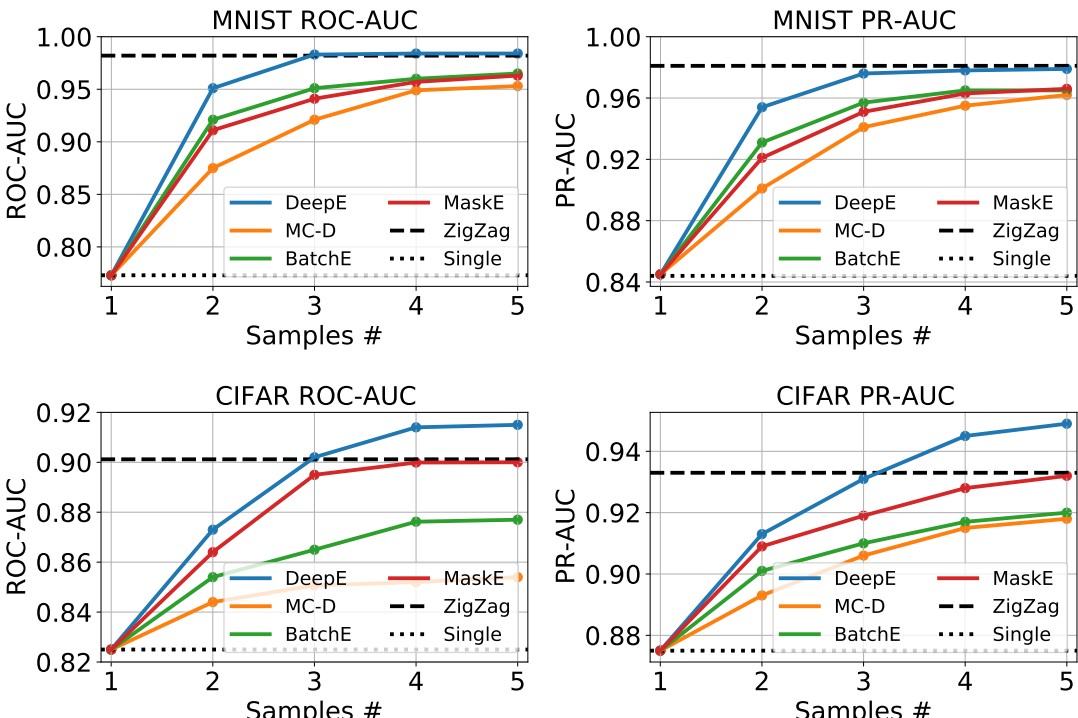

Figure 7: **OOD classification performance as a function of the number of samples.** The dashed line represents the performance of *ZigZag*, which is sampling-free.

with ReLU activations. This architecture yield good performance in terms of accuracy, outperforming the popular ordinal regression model CORAL (Cao et al., 2020) and matching other state-of-the-art approaches such as (Berg et al., 2021). As in the classification experiments described above, we use iCartoonFace (Zheng et al., 2020) dataset as out-of-distribution data. It comprises about 400k images of cartoon and anime character faces whose pixel statistics are different from those of the UTKFace while exhibiting a semantic similarity. As before, we train our model on the UTKFace training set and use uncertainty to distinguish UTKFace test set images from iCartoonFace ones.

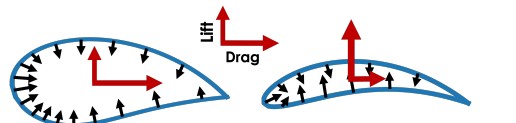

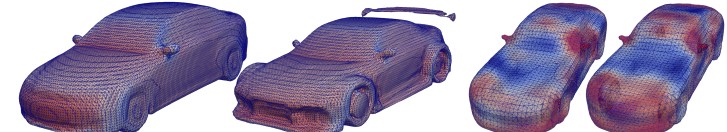

Figure 8: **Airfoil Samples.** Training and testing profiles (**left**) have a reasonable level of aerodynamics, whereas out-of-distribution samples (**right**) include only top-notch, but rare, shapes in terms of the lift-to-drag ratio. The black arrows represent pressures while the red lines depict overall lift and drag.

Figure 9: **Left: Pressure values.** The car dataset comprises many regular vehicles (left) and a few streamlined ones (right), which we treat as being out-of-distribution. Red and blue denote high and low pressures respectively. **Right: Pressure differences.** Large differences in predicted pressure are shown in red and low ones in blue. Ensembles (left) and ZigZag (right) yield the same pattern, with large values mostly in high-curvature areas.

**Predicting Lift-to-Drag Ratios.** Our method is generic and can operate with any kind of data. To demonstrate this, we collected a dataset of $2k$ wing profiles such as those of Fig. 8 by sampling the widely used NACA parameters (Jacobs & Sherman, 1937). We then ran the popular XFoil simulator (Drela, 1989) to compute the pressure distribution along each profile and estimate its lift-to-drag coefficient, a key measure of aerodynamics performance. The resulting dataset consists of wing profiles $\mathbf{x}_i$ represented by a set of 2D nodes and the corresponding scalar lift-to-drag coefficient $\mathbf{y}_i$ for $1 \le i \le 2000$.

We took the 5% of top-performing shapes in terms of lift-to-drag ratio to be the out-of-distribution samples. We took 80% of the remaining 95% as our training set and the rest as our test set. Hence, training and testing shapes span lift-to-drag values from 0 to 60, whereas everything beyond that is considered to be OOD and therefore not used for training purposes. We then trained a Graph Neural Network (*GNN*) that consists of 25 GMM (Monti et al., 2017) layers with ELU nonlinearities (Clevert et al., 2015) and skip connections (He et al., 2016) to predict lift-to-drag $\mathbf{y}_i$ from profile $\mathbf{y}_i$ for all $i$ in the training set, as in (Remelli et al., 2020; Durasov et al., 2021b). See additional details in the appendix.

**Predicting the Drag Coefficient of a Car.** We performed a similar experiment on 3D car models from a subset of the ShapeNet dataset (Chang et al., 2015) that features car meshes that are suitable for CFD simulation. We used the same experimental protocol as for the wings except for relying on OpenFOAM (Jasak et al., 2007) to estimate the drag coefficients and a more sophisticated network to predict it from the triangulated 3D meshes representing the cars, which we also describe in the supplementary material.

|           | Single | MC-D | DeepE | *ZigZag* |
|-----------|--------|------|-------|----------|
| MAE (↓)   | 22.9   | 20.3 | **17.9** | 19.2  |
| rAULC (↑) | 0.55   | 0.62 | 0.68  | **0.69** |
| ROC-AUC (↑) | 0.64 | 0.74 | 0.83  | **0.84** |
| PR-AUC (↑) | 0.63  | 0.77 | **0.84** | 0.82  |

Table 3: **Pressure Prediction.** Accuracy and calibration are computed for individual predictions for each node of the mesh. AUC metrics are computed using averaged uncertainty of the mesh and the same data splits as for the drag prediction task.

| **MNIST** | MC-D | MaskE | BatchE | DeepE | ZigZag |
|-----------|------|-------|--------|-------|--------|
| ROC-AUC   | 0.875 | 0.911 | 0.921 | 0.951 | **0.982** |
| PR-AUC    | 0.901 | 0.921 | 0.931 | 0.954 | **0.981** |
| **CIFAR** | MC-D | MaskE | BatchE | DeepE | ZigZag |
| ROC-AUC   | 0.844 | 0.864 | 0.854 | 0.873 | **0.901** |
| PR-AUC    | 0.893 | 0.909 | 0.901 | 0.913 | **0.933** |

Table 4: **Two-sample inference evaluation for sampling-based methods.** The tables present ROC-AUC and PR-AUC scores for the MNIST and CIFAR datasets, with the highest scores in each category highlighted. When constrained to the same computational requirements, *ZigZag* outperforms the other approaches.

To experiment with a higher-dimensional task, we used the same data to train a network to predict not only the drag but also a pressure value at each vertex of a car, as shown at the top of Fig. 9. We used the same train-test split as before along with a modified version of the network we used for drag prediction in which we replaced some convolutional layers by transformer layers (Shi et al., 2020), as explained in the supplementary material. As shown at the bottom of Fig. 9, *ZigZag* yields per-node uncertainties very similar to those of Deep Ensembles. The most uncertain regions are high curvature parts where pressure changes rapidly. This is reflected by the quantitive results of Tab. 3 that, once again, show *ZigZag* and Deep Ensembles performing similarly.

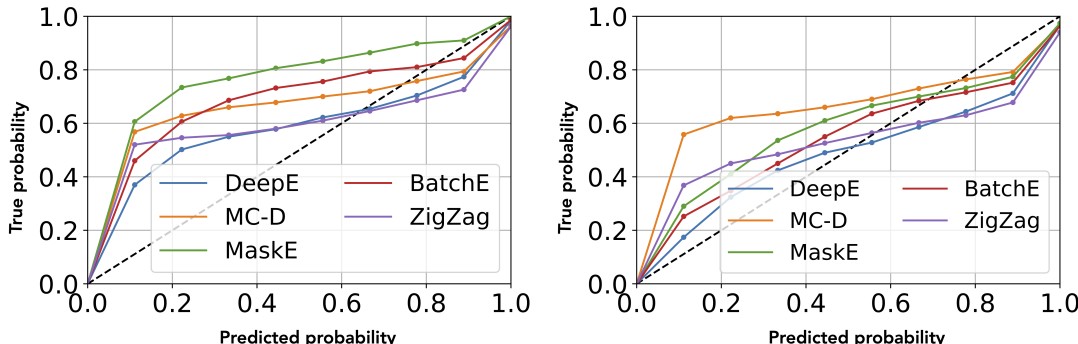

Figure 10: **Actual error vs uncertainty estimate** for **Airfoils** (left) and **Cars** (right) test sets. The $y = x$ curve denotes perfect calibration. ZigZag behaves like ensembles and better than the others.

The rAULC metric we have been reporting aggregates information on how well calibrated the uncertainty estimates are. To better visualize the calibration characteristics of the various methods, we provide actual calibration curves for the airfoil and car aerodynamics regression tasks. In Fig. 10, we plot the cumulative

probability of our estimation error being between zero and its maximum value against the cumulative probability of our uncertainty estimate similarly being between its minimum and maximum values, as in (Kendall & Gal, 2017). An ideal result would follow the diagonal. *ZigZag* and *DeepE* produce the results closest to that, which is consistent with the rAULC calibration results of Tab. 3.

### 4.5 Ablation study

**Two-sample Evaluation for Sampling-based Approaches.** A major strength of our method is its inference speed: It only requires two feed-forward passes. Here we conducted additional evaluations where we limited sampling-based approaches to two samples to compare how each scheme performs under the same inference cost in terms of uncertainty measurement. In Tab. 4, we present uncertainty estimates for MNIST (Top) and CIFAR (Bottom), running all sampling-based methods with an inference cost equal to ZigZag's — 2x compared to a Single model. Under these computational cost constraints, our method offers superior uncertainty estimates. For the ImageNet experiments, we evaluated two-sample ensembles, achieving 81% rAULC, 53% ROC-AUC, and 16% PR-AUC, metrics which are surpassed by *ZigZag*, as shown in Tab. 1.

**Additional Baselines.** For completeness sake, we evaluated two additional baselines that can only be run on a subset of all our testing examples: Temperature Scaling (*TempS*) for classification (Guo et al., 2017) and using a separate model for variance prediction in regression (*TwoM*) (Nix & Weigend, 1994). As shown in Tab. 6, temperature scaling improves calibration on the MNIST dataset, as shown by the rAULC metric, but is less good at evaluating epistemic uncertainty and detecting OOD samples. Tab. 5 includes the *TwoM* baseline consisting of a regression network and a separate model predicting the output variance, compared with our method in aerodynamic cases: airfoils and cars. This approach improves uncertainty metrics over a single model but does not match our method's performance, despite similar computational complexity. Further analysis can be found in Sections A.3 to A.5 of the supplementary material.

Table 5: **Two-Model Variance Prediction:** *TwoM* improves uncertainty metrics over single models but falls short of our method, with comparable computational complexity.

| **Airfoils** | Single | TwoM | DeepE | ZigZag | **Cars** | Single | TwoM | DeepE | ZigZag |
|---|---|---|---|---|---|---|---|---|---|
| MAE | 3.18 | 3.20 | **3.03** | 3.10 | MAE | 0.121 | 0.122 | **0.101** | 0.112 |
| rAULC | 0.008 | 0.027 | 0.062 | **0.068** | rAULC | 0.03 | 0.04 | **0.10** | 0.07 |
| Size | 1x | 2x | 5x | 1x | Size | 1x | 2x | 5x | 1x |
| Inf. Time | 1x | 2x | 5x | 2x | Inf. Time | 1x | 2x | 5x | 2x |
| Time | 1x | 1x | 5x | 1.2x | Time | 1x | 1x | 5x | 1.2 |
| ROC-AUC | 0.690 | 0.878 | 0.972 | **0.992** | ROC-AUC | 0.755 | 0.849 | 0.954 | **0.956** |
| PR-AUC | 0.681 | 0.882 | 0.955 | **0.987** | PR-AUC | 0.534 | 0.811 | 0.941 | **0.974** |

| | Accuracy (↑) | rAULC (↑) | Size | Inf. Time | Time | ROC-AUC (↑) | PR-AUC (↑) |
|---|---|---|---|---|---|---|---|
| TempS | 0.980 | 0.857 | 1x | 1x | 1x | 0.768 | 0.836 |
| *ZigZag* | **0.982** | **0.961** | 1x | 2x | 1x | **0.982** | **0.981** |

Table 6: **Temperature Scaling Evaluation.** Although temperature scaling significantly boosts the model's calibration, it does not improve and might slightly impair the model's capacity to detect OOD samples.

## 5 Conclusion

We have proposed an approach to estimating uncertainty that only requires performing a minor change in the first layer of network to accept an additional argument that may be either blank or the result of that prediction. Training the network to yield the same result in both cases enables us to estimate the uncertainty in a principled way and at low computational cost. This approach is applicable to any practical architecture and requires minimal modifications. It is easy to deploy, generic, and delivers results on par with ensembles at a much reduced training budget.

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

# A  Appendices

In this appendix, we describe the calibration metrics we use and provide additional details about the baselines, training setups, and hyper-parameters used in the experimental section.

## A.1  Calibration Metrics

In this section, we will describe metrics used for calibration evaluation both for classification and regression tasks. Typical calibration metrics such as *Expected Calibration Error* (ECE) (Guo et al., 2017) require uncertainties to be express in probabilistic form, which is not the case for many single-shot uncertainty methods. Therefore, unified calibration should be utilized that suits all of the available methods. One of such metrics is *Relative Area Under the Lift Curve* (rAULC) (Postels et al., 2022) which is based on the *Area Under the Lift Curve* (Vuk & Curk, 2006).

This metric is obtained by ordering the samples according to increasing uncertainty and calculating the accuracy of all samples with an uncertainty value smaller than a particular quantile. More formally, producing uncertainty value $\mathbf{u}_i$ for every sample in our evaluation set, we also generate an array of uncertainty quantiles $\mathbf{q}_j \in [0, 1], i \in [1, ..., S]$, with the quantile step equal to $1/S$. Iterating over quantiles $\mathbf{q}_j$, we compute the performance of our model $F(\mathbf{q}_j)$ using only samples for which uncertainty is less than this quantile. Finally, following notation from Postels et al. (2022) we compute the AULC metric as

$$AULC = -1 + \sum_{S}^{j=1} \frac{1}{S} \frac{F(\mathbf{q}_j)}{F_R(\mathbf{q}_j)},$$

where $F_R(\mathbf{q}_j)$ represents performance for baseline uncertainty that corresponds to random ordering. Further, in order to compute rAULC we divide AULC with the value of AULC produced by ideal (optimal) uncertainty model that perfectly orders all of the samples in order of increasing error. Following Postels et al. (2022), we use accuracy as $F(\mathbf{q}_j)$ for classification. Similarly, we extend AULC and rAULC to regression tasks via using as $F(\mathbf{q}_j)$ an inverse of *Mean Absolute Error* (MAE) computed for samples with uncertainties less then $\mathbf{q}_j$.

## A.2  Training Details and Baselines

**Synthetic Regression**  For our synthetic regression experiments, we use the architecture that consists of 6 linear blocks, ELU (Clevert et al., 2015) activations, BatchNorms (Ioffe & Szegedy, 2015) and skip-connections (He et al., 2016). We train the model for 4000 epochs using Adam (Kingma & Ba, 2015) optimizer with $10^{-2}$ learning rate and *mean squared error* loss. For *Single* baseline, we utilize loss from (Kendall & Gal, 2017) to enable uncertainty estimation. *Deep Ensembles* baseline uses 5 trained single models to extract mean and variance from predictions. For *MC-Dropout*, we apply dropout with 0.2 dropout rate to the last 2 linear layers and sample 5 different predictions during inference. Lastly, for *ZigZag* we extend the first layer of single model to take two inputs and train it the same way as original model.

**Synthetic Classification**  For synthetic classification experiments, we adopt simple feed-forward neural network that comprise of 10 linear layers with ELU activation and skip-connections. As for regression, we apply Adam optimizer for 300 epochs and $10^{-2}$ learning rate. *Deep Ensembles* also consists of 5 models, *MC-Dropout* drops activations from the last two layers with 0.15 drop rate, *ZigZag* extends the first layer of the original model so it is able to process additional inputs.

**MNIST**  Model used for MNIST experiments consists of two convolutional layers with max polling followed by three linear layers with LeakyReLU activations. We also train this model using Adam optimizer for three epochs with $10^{-2}$ learning rate. *MC-Dropout, BatchEnsemble and Masksembles* are applied to the last two layers of the model with 0.2 drop rate for *MC-Dropout* and 1.5 scale factor for *Masksembles*. *VarProp* propagates variance through the last three linear layers as it was described in (Postels et al., 2019). *SNGP* applies Random Features (Rahimi & Recht, 2007) to the last layer of the model and Spectral

Normalization (Miyato et al., 2018) to the rest. *OC* extracts features after convolutional layers and train five small models that represent certificates.

**CIFAR**  For CIFAR experiments, we use DLA (Yu et al., 2018) network and adopt original training setup: network is trained with SGD optimizer with 0.9 momentum for 20 epochs with $10^{-1}$ learning rate and 10 more epochs with $10^{-2}$. As before, *MC-Dropout, BatchEnsemble and Masksembles* are applied to the last three layers of the model with 0.1 drop rate and 1.5 scale factor. Features for *OC* are taken after convolutional part of the model.

**ImageNet**  For ImageNet experiments, we use common ResNet50 (He et al., 2016) architecture and follow original training procedure. *MC-Dropout, BatchEnsemble and Masksembles* are applied to the last five layers of the model with 0.1 drop rate, 2.0 scale factor, and each uses 5 samples during inference. *Varprop* propagates variance through the last five layers of the network. *OC* uses features from the penultimate layer and trains five small feed-forward networks for certificates.

**Age Prediction**  As an age predictor, we use common Resnet (He et al., 2016) backbone followed by five linear layers with LeakyReLU activations. As before, *MC-Dropout, BatchEnsemble and Masksembles* are applied to the last four layers of the model with 0.1 drop rate and 1.5 scale factor. *Varprop* propagates variance through the last five layers of the network. *OC* uses features from penultimate layer and trains five small feed-forward networks for certificates.

**Airfoils Lift-to-Drag**  Lift-to-Drag ratio is predicted with custom model that consists of twenty five GMM (Monti et al., 2017) layers, global max pooling and five linear layers with applied ReLU activations. The model is trained for 10 epochs with Adam optimizer and $10^{-3}$ learning rate. All of the uncertainty baselines follow the same setup described for age prediction experiments.

**Estimating Car Drag**  To predict drag associated to a triangulated 3D car, we utilize similar model to airfoil experiments but with increased capacity. Instead of twenty five GMM layers, we use thirty five and also apply skip-connections with ELU activations. Final model is being trained for 100 epochs with Adam optimizer and $10^{-3}$ learning rate. All of the uncertainty methods are applied to non-graphical part of the model – last five linear layers. As before, *MC-Dropout* uses 0.05 drop rate, *Masksembles* use 1.5 scale factor, *SNGP* applies Spectral Normalization to the last five layers and *OC* extract features right after max pooling layer.

For pressure prediction task, we modified original architecture and replaced GMM layers with Transformer layers (Shi et al., 2020) for more fine-grained predictions. In addition, we use GraphNorm (Cai et al., 2021) after each convolution for faster training and increase total size of the model to seventy layers. The model is being trained for 1500 epochs with Adam optimizer with $10^{-3}$ learning rate. Implemented uncertainty baselines are replicated from drag prediction experiments.

### A.3    Standard deviation of results

| | Metric | Single | MC-D | MaskE | BatchE | DeepE | ZigZag |
|---|---|---|---|---|---|---|---|
| **MNIST** | Accuracy (↑) | $0.980 \pm 0.002$ | $0.981 \pm 0.002$ | $0.989 \pm 0.004$ | $0.989 \pm 0.007$ | $0.990 \pm 0.005$ | $0.982 \pm 0.002$ |
| | rAULC (↑) | $0.712 \pm 0.008$ | $0.932 \pm 0.008$ | $0.929 \pm 0.017$ | $0.941 \pm 0.004$ | $0.958 \pm 0.004$ | $0.961 \pm 0.005$ |
| | ROC-AUC (↑) | $0.773 \pm 0.006$ | $0.953 \pm 0.011$ | $0.963 \pm 0.006$ | $0.965 \pm 0.003$ | $0.984 \pm 0.006$ | $0.982 \pm 0.007$ |
| | PR-AUC (↑) | $0.844 \pm 0.007$ | $0.962 \pm 0.010$ | $0.966 \pm 0.011$ | $0.965 \pm 0.003$ | $0.979 \pm 0.006$ | $0.981 \pm 0.007$ |
| **CIFAR** | Accuracy (↑) | $0.890 \pm 0.004$ | $0.909 \pm 0.001$ | $0.901 \pm 0.003$ | $0.911 \pm 0.003$ | $0.929 \pm 0.002$ | $0.928 \pm 0.003$ |
| | rAULC (↑) | $0.884 \pm 0.005$ | $0.889 \pm 0.004$ | $0.889 \pm 0.004$ | $0.884 \pm 0.005$ | $0.911 \pm 0.002$ | $0.897 \pm 0.004$ |
| | ROC-AUC (↑) | $0.825 \pm 0.017$ | $0.854 \pm 0.016$ | $0.900 \pm 0.002$ | $0.877 \pm 0.006$ | $0.915 \pm 0.004$ | $0.901 \pm 0.002$ |
| | PR-AUC (↑) | $0.875 \pm 0.018$ | $0.918 \pm 0.015$ | $0.931 \pm 0.002$ | $0.919 \pm 0.004$ | $0.949 \pm 0.004$ | $0.933 \pm 0.002$ |

Table 7: **Classification results on MNIST (Top) and CIFAR (Bottom) datasets.** Results for training and evaluation using three different random seeds, presented as averages with standard deviations included in the table.

All training and evaluation were conducted using three different random seeds, with the results then averaged. We report these results in two tables below (the top one for MNIST, the bottom for CIFAR), including standard deviation values for the best-performing baselines. These results show that ZigZag and Ensembles significantly exceed the performance of other baselines in terms of statistical significance.

### A.4 Split MNIST Evaluation

|  | Single | MC-D | MaskE | BatchE | DeepE | ZigZag |
|---|---|---|---|---|---|---|
| Accuracy (↑) | 0.992 | 0.993 | 0.992 | 0.990 | **0.998** | 0.993 |
| rAULC (↑) | 0.952 | 0.954 | 0.991 | 0.989 | **0.994** | 0.992 |
| ROC-AUC (↑) | 0.929 | 0.923 | 0.945 | 0.947 | 0.979 | **0.994** |
| PR-AUC (↑) | 0.923 | 0.918 | 0.945 | 0.947 | 0.977 | **0.991** |

Table 8: **Classification results on MNIST-S.** The best result in each category is highlighted in **bold**, while the second best is in **bold**, with most of these results attributed to ZigZag and DeepE. This indicates that while their performance is comparable, ZigZag has the advantage of requiring significantly less computation and memory. The high performance of *ZigZag* is consistent across MNIST and MNIST-S results.

In our experiments for the out-of-distribution (OOD) task in Sec.4.3, we employed standard setups from previous research(Malinin & Gales, 2018; Ciosek et al., 2019; Durasov et al., 2021a). For OOD detection in MNIST experiments, using FashionMNIST, which contains images markedly different from MNIST, is a common benchmark. To enhance our evaluation with a more challenging setup, we conducted additional MNIST experiments—referred to as MNIST-S—using digits 0-4 as in-distribution and 5-9 as OOD. The results, presented in Tab. 8, were obtained using the same architecture and training setups as the original MNIST experiments. In these experiments, both Ensembles and ZigZag demonstrated superior performance, compared to other baselines.

### A.5 AutoEncoder Baseline

As discussed in Sections 2 and 3, there is a strong connection between our method and autoencoder models. This makes the reconstruction error of an autoencoder a relevant baseline for comparison with our method. Uncertainty metrics for several datasets (MNIST, MNIST-S, CIFAR) have been included for this baseline below. For this approach, an autoencoder with a standard hourglass architecture was trained independently from the original classification model. The reconstruction error of the autoencoder is used as an uncertainty measure for an input $\mathbf{x}$. While this method shows promising results in moderately easy out-of-distribution (OOD) detection tasks, it struggles in more complex scenarios. Both in terms of uncertainty calibration and OOD detection, it underperforms, as demonstrated in Tab. 9, significantly lagging behind ZigZag and other baselines. The poor calibration of the reconstruction error for $\mathbf{x}$ arises from its exclusive focus on $\mathbf{x}$ during predictions, only estimating the likelihood of $\mathbf{x}$ and not the prediction $\mathbf{y}$. While suitable for OOD detection, it is inadequate for calibrated uncertainty predictions. Additionally, image reconstruction is more computationally intensive than classification, substantially adding to the computational burden of the approach. Therefore, the autoencoder was not deemed a viable baseline in other experiments, due to these calibration, computational, and *aleatoric* uncertainty issues.

|  | Accuracy (↑) | rAULC (↑) | Size | Inf. Time | Time | ROC-AUC (↑) | PR-AUC (↑) |
|---|---|---|---|---|---|---|---|
| MNIST | 0.98 | 0.35 | 1.5x | 3x | 2x | 0.94 | 0.96 |
| MNIST-S | 0.99 | 0.46 | 1.5x | 3x | 2x | 0.60 | 0.56 |
| CIFAR | 0.89 | 0.26 | 2x | 4x | 2x | 0.38 | 0.66 |

Table 9: **Autoencoder reconstruction error baseline.** The table shows results for an autoencoder baseline on MNIST, MNIST-S, and CIFAR datasets. Its limited calibration accuracy stems from focusing only on $\mathbf{x}$ and not $\mathbf{y}$. Though somewhat effective for OOD detection, it's more computationally costly than *ZigZag* due to the complexity of image reconstruction.

