# OpenReview forum: "ZigZag: Universal Sampling-free Uncertainty Estimation Through Two-Step Inference"
_TMLR — Accepted by TMLR_

### Review · Reviewer_t2aE · 2024-02-04

**Summary Of Contributions:**

The paper proposes ZigZag, a simple method for estimating out of distribution uncertainty, which by its generic formulation is applicable to different tasks (classification, regression) and different network architectures (dense nets, convolutional, graph nns). The experiments show that ZigZag has an advantage in terms of quality of uncertainty estimates over standard baselines such as Ensembles and Dropout, and additionally it is more resource efficient, requiring less memory and time at inference time.

**Audience:**

Yes

**Broader Impact Concerns:**

No ethical concerns.

**Claims And Evidence:**

Yes

**Requested Changes:**

Please see my notes above. I believe this is an interesting idea but there is lot that can be improved in the method presentation and analysis to make this a full ML method submission.

**Strengths And Weaknesses:**

Strengths:
- simple and general method applicable to different problem setups
- interesting datasets and applications in experiments beyond the standard image datasets ( including engineering problems such as airfoil and car shape optimisation)
- the experimental results look promising

Weaknesses:

Method:
- lack of method analysis - since the main contribution of the paper is a new method proposal, I would expect more analysis on the intuition such as, why should we expect this to work, what type of uncertainty does the method address - epistemic or aleatoric? Is the uncertainty related to the input data/features only ($\mathbf{x}$) or some uncertainty from the labels will be picked up?
- I believe your method is also sensitive to noise in the data, something that the authors should discuss more.
- What happens in the case when the error is already making prediction errors on some of the training examples, how does that affect ZigZag's behavior?

Experiments
- there is no standard error or deviation for the presented results, how many times were the experiments repeated? It is essential to know if the difference in performance is within margin of error with the rest of the methods
- the calibration results signal a need for improvement, seems like ZigZag is better than baselines but far from calibrated (close to the diagonal line)
- Why is the autoencoder as the most obvious baseline due to the similarity to ZigZags approach not included in experiments?
- In the image experiments, it is common to evaluate OOD performance in a more difficult setup , when the train and OOD split is from the same dataset but different classes (ex. MNIST 0-4 in distribution and 5-9 test, or OOD distribution)

Minor:
- Fig 2 is unnecessary

---

> ### Author Response · Authors · 2024-02-25
> **Authors Response to Reviewer t2aE**
>
> We thank the reviewer for their thorough review and instructive feedback. We highly appreciate that they find our approach general and effective. We have carefully addressed all concerns and suggestions, as discussed below.
>
> ---
>
> * **"Why should we expect this to work?"**
>
> We train the network so that the second feed-forward pass  reconstructs the second argument. As in an auto-encoder, we expect the reconstruction error to be low for in-distribution data and high for out-of-distribution data, thereby providing an estimation of uncertainty. More specifically, when we provide the network $\hat{\mathcal{M}}(\mathbf{x},\mathbf{0})$ with input $(\mathbf{x}, \mathbf{y})$, if the label $\mathbf{y}$ is close to the correct answer, then the difference between $\hat{\mathcal{M}}(\mathbf{x},\mathbf{0})$ and $\hat{\mathcal{M}}(\mathbf{x},\mathbf{y})$ is small because the network is trained to behave in this manner. However, if $\mathbf{y}$ is **not** close to the correct answer, then $(\mathbf{x}, \mathbf{y})$ represents something that the network has never encountered during training: a first argument $\mathbf{x}$ and a second argument $\mathbf{y}$ that is neither $\mathbf{0}$ nor the ground-truth. Essentially, this is an out-of-distribution sample for which the network, like most networks, can be expected to produce an unpredictable output. Hence, there is no reason for  $\hat{\mathcal{M}}(\mathbf{x},\mathbf{0})$ and $\hat{\mathcal{M}}(\mathbf{x},\mathbf{y})$ to be similar. In practice, as shown by Fig. 4, they are not, and we leverage this observation to quantify the model's uncertainty. In other words, there are two scenarios when reconstruction fails due to the OOD property: 1) when $(\mathbf{x}, \mathbf{y})$ is OOD because $\mathbf{x}$ is OOD, addressing *epistemic* uncertainty and OOD samples, 2) when $(\mathbf{x}, \mathbf{y})$ is OOD because $\mathbf{y}$ is OOD / errornous. In this case, the reconstruction issue is due to $\mathbf{y}$, our uncertainty measure is high, we cover *aleatoric* uncertainty connected to predicted target.
>
> * **"What type of uncertainty does the method address: epistemic or aleatoric?":**
>
> There are two main scenarios that can result in a large distance between the two estimates $\mathbf{y}_0$ and $\mathbf{y}_1$.  One denotes epistemic uncertainty and the other aleatoric uncertainty.
>
>
> - *Epistemic uncertainty.* When $\mathbf{x}$ is OOD, there is epistemic uncertainty and the reconstruction usually fails. This yields a $\mathbf{y}_0$ that is far from the ground-truth. Thus the pair $(\mathbf{x},\mathbf{y}_0)$ fed to the network for the second prediction $\mathbf{y}_1$ is also OOD and  $\mathbf{y}_1$ is likely to be very different from $\mathbf{y}_0$, as explained above.
>
> - *Aleatoric uncertaity.* When $\mathbf{x}$ is in in-distribution but the prediction $\mathbf{y}_0$ is nevertheless wrong, it is a consequence of aleatoric uncertainty. The pair $(\mathbf{x},\mathbf{y}_0)$ is again OOD because $\mathbf{y}_0$ is neither $\mathbf{0}$ not the ground-truth, resulting again in a large distance between the two predictions.
>
> Tables 1 and 2 of the main paper provide strong experimental evidence that this is indeed what happens: We use the *rAULC*, *ROC-*, and *PR-AUC* metrics to evaluate our uncertainty measure. *rAULC* is related to the calibration of the model [1], thus it can be regarded as a measure of *aleatoric* uncertainty. In contrast, *AUCs* gauges the model's ability to detect out-of-distribution samples [2], as high uncertainty is expected for them. This is typically associated with *epistemic* uncertainty.
>
> * **"Is the uncertainty related to the input data/features only ($\mathbf{x}$) or does it also include uncertainty from the labels?"**
>
> Both. The OOD experiments show that we capture uncertainty due the data/features whereas the scatter plots of Fig. 4 demonstrate that our uncertainty estimates correlate very strongly with the label's accuracy.
>
> * **"What happens in the case when the model is already making prediction errors on some of the training examples, how does that affect ZigZag's behavior?"**
>
>
> When the model incorrectly predicts $\tilde{\mathbf{y}} = \mathcal{M}(\mathbf{x}, \mathbf{0})$ for a training sample $\mathbf{x}$, the resulting reconstruction $\mathcal{M}(\mathbf{x}, \tilde{\mathbf{y}})$ tends to be erroneous as well and the estimated uncertainty is high. This is the same behavior we observe at inference time, as discussed above. This is precisely the behavior we aim for given inputs resulting in incorrect predictions. The model making an error on a training sample has little impact on training, as the second term of the loss function uses the actual target, not the predicted one, as input.
>
> [1] Postels, J., et al. 'Deterministic Epistemic Uncertainty.', ICML, 2022
>
> [2] Malinin, A., Gales, M. 'Predictive Uncertainty Estimation via Prior Networks.', NIPS, 2018.

---

> > ### Author Response · Authors · 2024-02-25
> > **Authors Response to Reviewer t2aE (Cont'd)**
> >
> > * **"I believe your method is also sensitive to noise in the data, something that the authors should discuss more"**
> >
> > As discussed above, noise in the training data may cause our model's prediction $\tilde{\mathbf{y}} = \mathcal{M}(\mathbf{x}, \mathbf{0})$ to be notably different from the desired target $\mathbf{y}$. This is the aleatoric uncertainty case discussed above and will result in an increased value of the uncertainty estimate.
> >
> > * **"Standard error or deviation for the presented results"**
> >
> > All training and evaluation was conducted using three different random seeds, and the results were then averaged. We report the results in two Tables below (top one for MNIST, bottom for CIFAR) with standard deviation values for the best-performing baselines. This supports the results in the paper: ZigZag and Ensembles outperform the other baselines by a statistically significant margin.
> >
> > |                        | Single | MC-D  | MaskE | BatchE | DeepE | ZigZag|
> > |------------------------|--------|-------|-------|--------|-------|---------|
> > | Accuracy ($\uparrow$)  | $0.980 \pm 0.002$  | $0.981 \pm 0.002$ | $0.989 \pm 0.004$  | $0.989\pm0.007$  | $0.990\pm0.005$ | $0.982\pm0.002$   |
> > | rAULC ($\uparrow$)     | $0.712\pm 0.008$  | $0.932\pm 0.008$ | $0.929\pm0.017$ | $0.941\pm0.004$  | $0.958\pm0.004$ | $0.961\pm0.005$   |
> > | ROC-AUC ($\uparrow$)   | $0.773\pm 0.006$  | $0.953\pm0.011$ | $0.963\pm0.006$ | $0.965\pm0.003$  | $0.984\pm0.006$ | $0.982\pm0.007$   |
> > | PR-AUC ($\uparrow$)    | $0.844\pm0.007$  | $0.962\pm0.010$ | $0.966\pm0.011$ | $0.965\pm0.003$  | $0.979\pm0.006$ | $0.981\pm0.007$   |
> >
> >
> > |                        | Single | MC-D  | MaskE | BatchE | DeepE | ZigZag|
> > |------------------------|--------|-------|-------|--------|-------|---------|
> > | Accuracy ($\uparrow$)  | $0.890\pm0.004$  | $0.909\pm0.001$ | $0.901\pm0.003$ | $0.911\pm0.003$  | $0.929\pm0.002$ | $0.928\pm0.003$   |
> > | rAULC ($\uparrow$)     | $0.884\pm0.005$  | $0.889\pm0.004$ | $0.889\pm0.004$ | $0.884\pm0.005$  | $0.911\pm0.002$ | $0.897\pm0.004$   |
> > | ROC-AUC ($\uparrow$)   | $0.825\pm0.017$  | $0.854\pm0.016$ | $0.900\pm0.002$ | $0.877\pm0.006$  | $0.915\pm0.004$ | $0.901\pm0.002$   |
> > | PR-AUC ($\uparrow$)    | $0.875\pm0.018$  | $0.918\pm0.015$ | $0.931\pm0.002$ | $0.919\pm0.004$  | $0.949\pm0.004$ | $0.933\pm0.002$   |
> >
> > * **"Calibration results signal a need for improvement"**
> >
> > Indeed, as noted in many studies, modern neural networks [3] and many uncertainty estimation methods [1] exhibit poor calibration on multiple tasks. Our experiments also reflect this trend. However, our calibration results also show that, although our approach does not achieve perfect calibration, it performs better than other methods and is comparable to ensembles, which are currently regarded as one of the best uncertainty methods.
> >
> > * **"In the image experiments, it is common to evaluate OOD performance in a more difficult setup , when the train and OOD splits are from the same dataset but different classes (ex. MNIST 0-4 is in-distribution and 5-9 is OOD)"**
> >
> > In our experiments for the out-of-distribution (OOD) task, we used standard setups from prior research [2, 4, 5]. Using FashionMNIST for OOD detection in MNIST experiments, where FashionMNIST contains images markedly different from MNIST, is a typical benchmark.
> >
> > Nevertheless, to address the reviewer's suggestion for a more complicated setup, we conducted additional MNIST experiments--referred to as MNIST-S below--with digits 0-4 as in-distribution and 5-9 as OOD. The results are presented in Table below. They were obtained using the same architecture and training setups as the original MNIST experiments. In these experiments, both Ensembles and ZigZag again demonstrated the best performance.
> >
> > |                        | Single | MC-D  | MaskE | BatchE | DeepE | ZigZag |
> > |------------------------|--------|-------|-------|--------|-------|---------|
> > | Accuracy ($\uparrow$)  | 0.992  | 0.993 | 0.992 | 0.990  | 0.998 | 0.993   |
> > | rAULC ($\uparrow$)     | 0.952  | 0.954 | 0.991 | 0.989  | 0.994 | 0.992   |
> > | ROC-AUC ($\uparrow$)   | 0.929  | 0.923 | 0.945 | 0.947  | 0.979 | 0.994   |
> > | PR-AUC ($\uparrow$)    | 0.923  | 0.918 | 0.945 | 0.947  | 0.977 | 0.991   |
> >
> > [3] Guo, C., Pleiss, G., Sun, Y., Weinberger, K. Q. 'On Calibration of Modern Neural Networks.' ICLR, 2017.
> >
> > [4] Ciosek, K., et al. 'Conservative Uncertainty Estimation by Fitting Prior Networks.', ICLR, 2019.
> >
> > [5] Durasov, N., Bagautdinov, T., Baque, P., Fua, P. 'Masksembles for Uncertainty Estimation.', CVPR, 2021.

---

> > > ### Author Response · Authors · 2024-02-25
> > > **Authors Response to Reviewer t2aE (Cont'd)**
> > >
> > > * **"Autoencoder as the most obvious baseline due to the similarity to ZigZag"**
> > >
> > > Indeed, as discussed in the related work and method sections, there is a deep connection between our method and autoencoder models, making the autoencoder a viable baseline for comparison. We have included uncertainty metrics for several datasets (MNIST, MNIST-S, CIFAR) for this baseline. An auto-encoder with a standard hourglass architecture was trained separately from the original classification model for this approach. For an input $\mathbf{x}$, the reconstruction error of the autoencoder is used as an uncertainty measure. This method shows promising performance in out-of-distribution (OOD) detection tasks in moderately easy setups but fails in more complex scenarios, both in terms of uncertainty calibration and OOD detection. It is significantly outperformed by ZigZag and other baselines. The poor calibration of the reconstruction error for $\mathbf{x}$ stems from its focus solely on $\mathbf{x}$ during predictions, estimating only the likelihood of $\mathbf{x}$ and not the prediction $\mathbf{y}$. This makes it suitable for OOD detection but not for calibrated uncertainty predictions. Furthermore, image reconstruction is a more computationally demanding task than classification, markedly increasing the computational overhead of the approach. Therefore, we did not consider the autoencoder as a baseline due to these calibration, computational, and *aleatoric* uncertainty issues.
> > >
> > > |                        | MNIST | MNIST-S | CIFAR |
> > > |------------------------|-------|---------|-------|
> > > | Accuracy ($\uparrow$)  | 0.98  | 0.99    | 0.89  |
> > > | rAULC ($\uparrow$)     | 0.35  | 0.46    | 0.26  |
> > > | Size  | 1.5x  | 1.5x    | 2x  |
> > > | Inf. Time    | 3x  | 3x    | 4x  |
> > > | Time    | 2x  | 2x    | 2x  |
> > > | ROC-AUC ($\uparrow$)   | 0.94  | 0.60    | 0.38  |
> > > | PR-AUC ($\uparrow$)    | 0.96  | 0.56    | 0.66  |

---

> > > > ### Comment · Reviewer_t2aE · 2024-03-11
> > > > **Thank you for your responses**
> > > >
> > > > I appreciate the detailed explanation on the intuition behind the proposed training scheme and I suggest the authors include these notes in an updated version of the manuscript.  The additional experiments run during the rebuttal period increase the confidence and strengthen the value of ZigZag compared to baselines. Interestingly, it seems like Deep Ensembles are still a strong competitor and given their widespread use I suggest the authors emphasize the difference in resource consumption between the two methods whenever possible. Also the AE baseline indeed seems to do well in terms of accuracy but poorly wrt other metrics as expected, however, including these results in the supplementary material will be helpful and instructional to TMLR readers and in general the community of researchers and practitioners of uncertainty estimation in deep models.

---

### Review · Reviewer_3uZz · 2024-02-09

**Summary Of Contributions:**

This work proposes ZigZag, a sampling free approach that is task agnostic and estimates the uncertainty with two inference steps. The method works by

- first modifying the given model M(x) to accept additional input of the same size as the prediction y, i.e., $\hat{M}(x,y)$

- during inference, it does two forward passes through this modified model, $y_0 =  \hat{M}(x,0)$, followed by $y_1 = \hat{M}(x,y_0)$

- finally, it estimates the uncertainty as the error $\| y_0 - y_1 \|$. Low distance between $y_0$ and $y_1$ corresponds to lower uncertainty.


	The network is trained to minimize $L(\hat{M}(x,0), y) + L(\hat{M}(x,y), y)$ such that the above inference procedure becomes valid as first component is traditional inference and the second one is knowing a prediction, does the outcome change. In this step, $L$ measures the loss between predicted and actual label $y$.

This work provides various experiments on benchmark datasets to compare the proposed scheme to various baselines in the literature.

**Audience:**

Yes

**Claims And Evidence:**

Yes

**Requested Changes:**

- In Fig.5 (c,d), what does the strength of the green and yellow color denote? For instance, in figure (c), there's presence of both yellow and greeen shade, does it have any significance? Also, one would expect the Deep Ensembles to have better uncertainty (but with higher computational cost) compared to ZigZag. It is unclear in this figure.

- For the regression and classification task in Sec.4.2, can you provide an estimate of how much is the computational cost difference between Deep Ensembles and ZigZag. Also, this section does not report the performance of newer approaches such as MaskEnsembles, SNCP, etc.

- Can you report the uncertainty estimate of the Deep Ensemble for the same budget as ZigZag (i.e. 2x instead of 5x in the Table 1)? This should represent the fact that under same inference cost, which of the two schemes yield better uncertainty measure.

- Have you used other distance measures instead of || y0 - y1 || to measure the uncertainty?

- Why is the inference cost of the Masked Ensembles similar as the Deep Ensembles?

- Can you provide a better understanding of the training procedure? Why minimize the sum of the two terms? Why not trade-off the two terms with some hyder-parameter that weighs the second component differently?

- Can you do a third pass over the network to improve the uncertainty estimate?

- Inference cost of the Deep Ensemble is proportional to the number of models in the ensemble. But, if the system allows for parallel computation, this cost can be reduced to a single forward pass. In contrast, ZigZag has sequential dependence of the previous pass as a result, this cost (2x) is rigid and cannot be improved if the system has parallelization in-built.

**Strengths And Weaknesses:**

**Strengths**
- ZigZag is a sampling free approach to measure uncertainty
- ZigZag is also task agnostic and can be applied to regression, classification and other tasks.
- It has a constant inference cost to measure the uncertainty, i.e., 2x the cost of a single forward pass of the network

**Weaknesses**
- ZigZag requires retraining the network and modifying the architecture to enable constant inference cost of uncertainty estimate. This is a non-trivial step and most practictioners would avoid due to the added complexity.
- Schemes such as Deep Ensembles also enable trading off inference cost for the levels of uncertainty estimate, i.e., higher inference cost would result in better uncertainty estimate. ZigZag lacks this ability.
- It is unclear why approaches such as Mask Ensembles have similar inference cost as the Deep Ensembles.
- ZigZag cannot utilize parallelization which Deep Ensembles can leverage (inference of the models in the ensembles can be done in parallel). ZigZag requires sequential dependence on the previous forward pass.

---

> ### Author Response · Authors · 2024-02-25
> **Authors Response to Reviewer 3uZz**
>
> We sincerely appreciate the valuable comments and feedback provided by the reviewer. We deeply appreciate the recognition of the positive aspects of our method and experimental justification. We have carefully addressed all of the concerns and suggestions raised, as discussed below.
>
> ---
>
> * **"Masksembles have similar inference cost as the Deep Ensembles."**
>
> In Tab. 1 and 2 of the experimental section, the inference time metric is taken to be how much time the model takes to compute uncertainties relative to single model without parallelization, as described in Section 4.1. In this scenario, Masksembles, along with other sampling-based approaches, such as BatchEnsemble or MC-Dropout, spends 1x time performing one sample. Hence, the total inference time when using 5 samples turns into 5x. So, while these approaches indeed reduce the total training time, they do not necessarily decrease the inference time, compared to ensembles.
>
> * **"ZigZag requires retraining the network and modifying the architecture to enable constant inference cost of uncertainty estimate. This is a non-trivial step and most practitioners would avoid due to the added complexity."**
>
> Both retraining the model and modifying the architecture are indeed needed and yields a most effective sampling-free approach. Note however, that the architecture modification is very simple and only affects the very first layer, as we have demonstrated in various use cases.  Furthermore, the one approach that can compete with us, Ensembles, requires training not one but several networks, which is even more demanding.
>
> * **"ZigZag cannot utilize parallelization which Deep Ensembles can leverage. If the system allows for parallel computation, the of cost of Ensembles can be reduced to a single forward pass."**
>
> We acknowledge this but the fact that, for ensembles, several networks have to be trained remains. Furthermore, there are setups, for example in robotics, where the parallelization is not an option due to limitations in GPU memory and energy consumption. In such resource-constrained scenarios, achieving similar results with less computation and memory usage is valuable. Finally, ZigZag could also be parallelized in batch inference scenarios: If we have an incoming stream of data batches $(\cdots, \mathbf{x}^{n}, \mathbf{x}^{n+1}, \cdots)$ and can parallelize two inferences of our model, then while processing the input $\mathbf{x}^{n+1}$, we can conduct one inference as $\tilde{\mathbf{f}}^{n+1}= \mathcal{M}(\mathbf{x}^{n+1}, \mathbf{0})$ (the first inference for the current batch) and the other inference as $\tilde{\mathbf{s}}^{n} = \mathcal{M}(\mathbf{x}^{n}, \tilde{\mathbf{f}}^{n})$ (the second inference for the previous batch). This approach allows our throughput to match that of parallelized ensembles. Additionally, if we can parallelize more than two inferences, our model will achieve even higher throughput than ensembles.
>
> * **"In Fig.5 (c,d), what does the strength of the green and yellow color denote?."**
>
>
> In these experiments, the network input $\mathbf{x}$ are the 2D coordinates of the point. In order to generate Figures 5-(a,b,c,d), we computed uncertainty for points $\mathbf{x} \in [-2, 3] \times [-2, 2]$ (where $\times$ means Cartesian product) for all of these four methods. Higher uncertainty for a point is color with a brigther color (yellow), while lower uncertainty is snow with dark color (violet).
>
> * **"For the regression and classification task in Section 4.2, can you provide an estimate of how much is the computational cost difference between Deep Ensembles and ZigZag. Also, this section does not report the performance of newer approaches such as Masksembles, etc."**
>
>
> Computational cost differences between Deep Ensembles and ZigZag in these experiments are similar to other classification experiments and we report them in the Table below. In these experiments, we wanted to show qualitatively and in visual way that some of the popular baselines can (Single and MC-Dropout) indeed can perform rather poorly. Also, we show that our method performs similar to Ensembles (which we considered as a strongest baseline): both methods show low uncertainty for inputs close to training samples (narrow yellow line for classification and red for regression), while demonstrating high uncertainty far from training set (broad yellow line for classification and red for regression). We will add several other sampling-based baselines for visualization in this section in the final version of the paper and also report numbers for these methods.
>
> |             | Single | MC-D | MaskE | BatchE | DeepE | ZigZag |
> |-------------|--------|------|-------|--------|-------|---------|
> | Size        | 1x     | 1x   | 1x    | 1x     | 5x    | 1x      |
> | Inf. Time   | 1x     | 5x   | 5x    | 5x     | 5x    | 2x      |
> | Time        | 1x     | 1x   | 1x    | 1x     | 1x    | 1x      |

---

> > ### Author Response · Authors · 2024-02-25
> > **Authors Response to Reviewer 3uZz (Cont'd)**
> >
> > * **"Can you report the uncertainty estimate of the Deep Ensemble for the same budget as ZigZag (i.e. 2x instead of 5x in the Table 1)? This should represent the fact that under same inference cost, which of the two schemes yield better uncertainty measure."**
> >
> > In the two tables below, we quantify the quality of the uncertainty estimates for MNIST (top table) and CIFAR (bottom table), for which we run all of the sampling-based method with inference cost equal to ZigZag's one -- 2x compared to Single model. In such a setup where the computational cost is restricted, our method delivers the best uncertainty estimates.
> >
> > |                     | MC-D  | MaskE | BatchE | DeepE | ZigZag   |
> > |---------------------|-------|-------|--------|-------|-----------|
> > | ROC-AUC ($\uparrow$) | 0.875 | 0.911 | 0.921  | 0.951 | **0.982** |
> > | PR-AUC ($\uparrow$) | 0.901 | 0.921 | 0.931  | 0.954 | **0.981** |
> >
> > |                     | MC-D  | MaskE | BatchE | DeepE | ZigZag   |
> > |---------------------|-------|-------|--------|-------|-----------|
> > | ROC-AUC ($\uparrow$) | 0.844 | 0.864 | 0.854  | 0.873 | **0.901** |
> > | PR-AUC ($\uparrow$) | 0.893 | 0.909 | 0.901  | 0.913 | **0.933** |
> >
> > * **"Can you provide a better understanding of the training procedure? Why minimize the sum of the two terms? Why not trade-off the two terms with some hyper-parameter that weighs the second component differently?"**
> >
> > Choosing this specific loss function was dictated by the fact that both inferences of the model are equally important for the uncertainty estimation. The first term of our loss ensures that the model provides the most accurate predictions, the second term ensures that reconstruction given an initial estimate is accurate. Too high a weight on this second term could hurt the model's performance, while too much weight on the the first term compromises uncertainty estimation. Thus, a balance between the two terms is essential. In the Table below, we report the performance and uncertainty quality metrics for MNIST dataset, for experiments where we changed weight parameters $p$ for the loss $(1-p) \cdot \mathcal{L} (\hat{\mathcal{M}}(\mathbf{x},\mathbf{0}),\mathbf{y}) +  p \cdot \mathcal{L} (\hat{\mathcal{M}}(\mathbf{x},\mathbf{y}),\mathbf{y}) $. Higher values of $p$ deteriorate accuracy of the model, while too low or too high values of $p$ reduce uncertainty quality. The best performance is obtained when the two terms are given the same weight.
> >
> > |                     | $p=0.0$ | $p=0.25$ | $p=0.5$       | $p=0.75$ | $p=1.0$ |
> > |---------------------|---------|----------|---------------|----------|---------|
> > | Accuracy ($\uparrow$) | 0.980   | 0.980    | **0.980**     | 0.821    | 0.152   |
> > | ROC-AUC ($\uparrow$) | 0.402   | 0.954    | **0.982**     | 0.968    | 0.563   |
> >
> > * **"Can you do a third pass over the network to improve the uncertainty estimate?"**
> >
> > We have tried feeding the second prediction back as an input. However, in our experiments, we observed no significant enhancements in uncertainty quality by doing this. For example, for our MNIST experiments by running third inference the AUCs metrics changed as ROC = $0.982 \rightarrow 0.9822$ and PR = $ 0.981 \rightarrow 0.9811$. Consequently, we decided not to increase the computational complexity of our methods by addiding additional feed-forward passes.
> >
> > * **"Have you used other distance measures instead of $|y_{0} - y_{1}|$ to measure the uncertainty?"**
> >
> > We have used other distance measures for our uncertainty computations, such as the $\ell_2$ norm, but didn't observe significant difference in uncertainty quality.

---

> > > ### Comment · Reviewer_3uZz · 2024-04-02
> > > **Response to author rebuttal**
> > >
> > > Thanks for the detailed response to my concerns and clarifying the ZigZag approach.

---

### Review · Reviewer_9k95 · 2024-02-19

**Summary Of Contributions:**

This paper proposes Zig-Zag, a sampling free approach to estimate uncertainty in neural networks. Experiments are performed on a variety of vision-related classification and regression tasks (MNIST, CIFAR, Imagenet, some image regression tasks, and a shapeNet based task).

Zig-Zag's representation of uncertainty is the distance between an augmented model that takes the estimated response, yhat, as input and the same model which takes a vector of zeros as input. Training is done by giving the model the true response, y as input and minimizing that as part of the loss.

**Audience:**

Yes

**Claims And Evidence:**

No

**Requested Changes:**

Some clarifications would be helpful:

- [high] Is "single" model from Kendall and Gal a standard model trained with their uncertainty based loss functions (Eq 12 for classification and Eq 8 in regression?

- [medium / high] A missing baseline to me is basically the same approach as Kendall and Gal but training the variance model separately. That is for regression, training a model to predict the mean, and then another model to predict the variance (original reference, [Nix & Weigend 1994](https://ieeexplore.ieee.org/document/374138). This baseline also has two forwards passes of costs.

- Another baseline, which could unlock some interesting fine-tuning based experiments with large models actually, is to train your loss function separately as well. Then, you could have the base model producing its own prediction yhat and using that as input. Architectures would not need to be the same either.

- [medium / high] A final missing baseline that is single pass is [temperature scaling](https://arxiv.org/abs/1706.04599), which uses a validation set to readjust the probabilities. It has the advantage of being single-pass at test time, and needs only a limited amount of training data to fix the calibration issues.



- [high] to make my point more clear above, I am a bit concerned that the trained model would be too prone to overfitting. One can see this in the linear regression setting where $\hat X = cat(X, y)$. Then, a trivial solution (and what becomes the MSE minimizing solution) is to set \beta_y = 1 and then the other predictors from X equal to zero. Only a two stage approach like your inference method would tend to get around this issue. I could see that the approach works because the model doesn't seem to have enough capacity for some reason but the theoretical underpinning here is still the same.

      - It would be nice to look at the code to see what's going on under the hood here.


- [high] How is the "variance" / "standard deviation" estimated in regression? I see it for classification (where a normalized score is fine), but it's not immediately clear to me how one gets from ||M(x, 0) - M(x, \hat y)|| to \hat \sigma(x) in Figure 6 for example. This actually seems critical for the success / failure of the whole approach.

*Presentation Comments*

[medium] show Figure 7 for Imagenet. I'm curious to know if 2-3 samples of ensembles are highly performant there as well.

[low] Label the axes for Figure 10. The caption does a good job but this is a presentation issue. Also, maybe make the dots there larger too.

[medium] blue / black shading seems somewhat inconsistent especially in Table 3. I'd suggest instead using bold for best only and italics for second best throughout.

[low] use alpha shading less than 1 in Figure 4 so that most of the points can be seen.

[low] end of section 4.3: "However, it then needs at least double..." Actually it needs quadruple the memory for the forwards pass ;) Would also suggest writing out 4 or 5 and 2 and 3 there.

[low] Don't capitalize ensembles throughout.

Section 2: not bayesian networks, but Bayesian neural networks.

Weller & jebara 2014 reference is clearly mis-cited.

**Strengths And Weaknesses:**

Overall, the paper seems well written and is clear.

Strengths:

- The writing and experiments are clearly presented. Overall, this paper is quite readable.

- The method seems to work about as well as ensembles with five samples, with a pretty clear set of benchmarks.

- thank you for the clear description of inference time and the tradeoffs therein.

Weaknesses:

- A priori, I am a bit theoretically concerned about passing the true response, y, as input into the loss function. This seems like it would make training unstable as the model would learn to rely on the response to predict the response.

- For a method that claims "minimal changes", adding in a response -> input embedding layer and then two forwards passes is somewhat non-trivial, although I agree less than many other approaches. Two forwards passes in the training procedure is also a non-trivial training slowdown (if i remember correctly forwards passes are 3x faster than backwards so you add about 25% into the total training cost).

- The updated loss function may further require additional hyperparameter sweeping to adjust the weight of the secondary loss also increasing complexity.

- There seems unlikely to be further reliability guarantees other than practicality. With Bayesian approaches, you get some amount of guarantee under much more strenuous conditions, while held-out set approaches like temperature scaling and conformal prediction also have some semblance of guarantees.

---

> ### Author Response · Authors · 2024-02-25
> **Authors Response to Reviewer 9k95**
>
> We are grateful to the reviewer for their detailed review and valuable suggestions regarding our paper. We also deeply appreciate their recognition of  of the clarity in our writing and the thoroughness of our experiments. We have carefully addressed all of the concerns and suggestions raised, as discussed below.
>
> ---
>
> * **"I am a bit concerned about passing the true response, y, as input into the loss function and that the trained model would be too prone to overfitting, since ideal optimization should result in inputs copying."**
>
> This scenario is indeed realistic, yet not problematic. As in an auto-encoder, a model can effectively memorize all samples of a specific distribution, though its performance declines for out-of-distribution samples. We exploit this very property to derive an uncertainty estimation. Moreover, we observe that the extent of the performance drop correlates with how far the samples deviate from the training distribution, which is indicative of uncertainty.
> In general, the risk of overfitting always exists but *only* for samples belonging within same statistical distribution as those in the training set. For example, if we train a simple MLP to be the identity function for all $x \in [-1, 1]$, there is no guarantee that it will still be the identity outside of that interval because deep networks are notoriously bad at extrapolation [1, 2].
>
> More specifically, when we provide the network $\hat{\mathcal{M}}(\mathbf{x},\mathbf{0})$ with input $(\mathbf{x}, \mathbf{y})$, if the label $\mathbf{y}$ is close to the correct answer, then the difference between $\hat{\mathcal{M}}(\mathbf{x},\mathbf{0})$ and $\hat{\mathcal{M}}(\mathbf{x},\mathbf{y})$ is small because the network is trained to behave in this manner. However, if $\mathbf{y}$ is **not** close to the correct answer, then $(\mathbf{x}, \mathbf{y})$ represents something that the network has never encountered during training: a first argument $\mathbf{x}$ and a second argument $\mathbf{y}$ that is neither $\mathbf{0}$ nor the ground-truth. Essentially, this is an out-of-distribution sample for which the network, like most networks, can be expected to produce an unpredictable output. Hence, there is no reason for  $\hat{\mathcal{M}}(\mathbf{x},\mathbf{0})$ and $\hat{\mathcal{M}}(\mathbf{x},\mathbf{y})$ to be similar. In practice, as shown by Fig. 4, they are not, and we leverage this observation to quantify the model's uncertainty.
>
> * **"For a method that claims "minimal changes", adding in a response to input embedding layer and then two forwards passes is somewhat non-trivial, although I agree less than many other approaches. Two forwards passes in the training procedure is also a non-trivial training slowdown (about 25\% into the total training cost)."**
>
> We acknowledge this but note that  our computational overhead is still lower than for many of the alternatives, with performance on par with Deep Ensembles that is significantly more expensive. Our "minimal changes" claim is relative to other sampling-free approaches: First, the architectural change is straightforward, adding just an extra feature/channel in the convolution of the very first layer. Second the loss function incorporates an easy-to-compute second term.
>
> * **"The updated loss function may further require additional hyperparameter sweeping to adjust the weight of the secondary loss also increasing complexity."**
>
> That is true.  However, an equal weighting of both terms results in the best performance as shown in the table below.
>
> |                     | $p=0.0$ | $p=0.25$ | $p=0.5$       | $p=0.75$ | $p=1.0$ |
> |---------------------|---------|----------|---------------|----------|---------|
> | Accuracy ($\uparrow$) | 0.980   | 0.980    | **0.980**     | 0.821    | 0.152   |
> | ROC-AUC ($\uparrow$) | 0.402   | 0.954    | **0.982**     | 0.968    | 0.563   |
>
> This stems from the fact that both inferences of the model are crucial for effective uncertainty estimation. The first term of our loss ensures that the model provides the most accurate predictions, the second term ensures that reconstruction given an initial estimate is accurate. Too high a weight on this second term could hurt the model's performance, while too much weight on the the first term compromises uncertainty estimation. Thus, a balance between the two terms is essential. In the Table above, we report the performance and uncertainty quality metrics for MNIST dataset, for experiments where we changed weight parameters $p$ for the loss $(1-p) \cdot \mathcal{L} (\hat{\mathcal{M}}(\mathbf{x},\mathbf{0}),\mathbf{y}) +  p \cdot \mathcal{L} (\hat{\mathcal{M}}(\mathbf{x},\mathbf{y}),\mathbf{y}) $. Higher values of $p$ degrade the model accuracy, while too low or too high values of $p$ reduce uncertainty quality.

---

> > ### Author Response · Authors · 2024-02-25
> > **Authors Response to Reviewer 9k95 (Cont'd)**
> >
> > * **"There seems unlikely to be further reliability guarantees other than practicality."**
> >
> > True. However, as discussed extensively in our response to the first question of reviewer t2AE, there is a strong rationale for doing what we do, which is borne out by our empirical data. With respect to comparing with methods that do offer guarantees, our experience and findings, along with those in several other papers [3, 4, 5], suggest that such methods provide solid guarantees in simpler cases but often may not perform as well in more complex scenarios involving neural networks. Therefore, while we value theoretical underpinnings, our paper prioritizes practical applicability and focuses more on addressing real-world challenges effectively.
> >
> > * **"Is "Single" model from Kendall and Gal a standard model trained with their uncertainty based loss functions (Eq. 12 for classification and Eq. 8 in regression?)"**
> >
> > For both classification and regression tasks, we followed the setup described in the original works. For regression tasks, we used a standard model trained with the uncertainty loss as outlined by [Kendall \& Gal]. For classification experiments, we applied the common measure of entropy of the predictive distribution, as detailed in [6, 7, 8]. This method of computing the uncertainty measure for a single model has shown [9] to be one of the best for a single network. We will clarify that in the final version of the paper.
> >
> > * **"A missing baseline to me is basically the same approach as (Kendall \& Gal) but training the variance model separately."**
> >
> > This is a good point and we have tried it. In the table below, *TwoModel* refers to the baseline proposed by the reviewer, which we compared to ours in the aerodynamics cases: airfoils lift-to-drag (top table) and drag for cars (bottom table) prediction. For the second model predicting uncertainty in the form of log-variance $\log{\sigma^2}$, we utilized the same architecture and trained both models simultaneously with an uncertainty loss. As shown, this method does provide some improvement in uncertainty metrics compared to the single model baseline. However, it doesn't quite match our method in performance, despite having similar computational complexity. Additionally, *TwoModel* is tailored for regression tasks and lacks a clear extension to classification problems, which is one of the challenges in modern uncertainty methods discussed in our related work section. In contrast, our approach is applicable to both classification and regression tasks.
> >
> > |             | Single | TwoModel | DeepE | ZigZag |
> > |-------------|--------|----------|-------|--------|
> > | MAE ($\downarrow$) | 3.18  | 3.20     | 3.03  | 3.10   |
> > | rAULC ($\uparrow$) | 0.008 | 0.027    | 0.062 | 0.068  |
> > | Size        | 1x     | 2x       | 5x    | 1x     |
> > | Inf. Time   | 1x     | 2x       | 5x    | 2x     |
> > | Time        | 1x     | 1x       | 5x    | 1.2    |
> > | ROC-AUC ($\uparrow$) | 0.690 | 0.878    | 0.972 | 0.992  |
> > | PR-AUC ($\uparrow$) | 0.681 | 0.882    | 0.955 | 0.987  |
> >
> > |             | Single | TwoModel | DeepE | ZigZag |
> > |-------------|--------|----------|-------|--------|
> > | MAE ($\downarrow$) | 0.121 | 0.122    | 0.101 | 0.112  |
> > | rAULC ($\uparrow$) | 0.03  | 0.04     | 0.10  | 0.07   |
> > | Size        | 1x     | 2x       | 5x    | 1x     |
> > | Inf. Time   | 1x     | 2x       | 5x    | 2x     |
> > | Time        | 1x     | 1x       | 5x    | 1.2    |
> > | ROC-AUC ($\uparrow$) | 0.755 | 0.849    | 0.954 | 0.956  |
> > | PR-AUC ($\uparrow$) | 0.534 | 0.811    | 0.941 | 0.974  |
> >
> >
> >
> > * **"Another baseline, which could unlock some interesting fine-tuning based experiments with large models actually, is to train your loss function separately as well."**
> >
> > This is an interesting idea that we will pursue in future work. Unfortunately, training large models would be hard in the time we have to provide this rebuttal. Furthermore, a potential drawback of the proposed baseline is the larger overall size of the models being trained, resulting in higher computational complexity and longer training times. In our method, we deliberately focused on unifying these two models, thus allowing for two predictions to be made using a single, smaller model.

---

> > > ### Author Response · Authors · 2024-02-25
> > > **Authors Response to Reviewer 9k95 (Cont'd)**
> > >
> > > * **"A final missing baseline that is single pass is temperature scaling, which uses a validation set to readjust the probabilities."**
> > >
> > > Temperature scaling, while effective in addressing the widely-recognized calibration issues of modern neural networks, has been observed to underperform in evaluating epistemic uncertainty and detecting Out-of-Distribution (OOD) samples or distribution shifts, as noted by [4, 10]. This stems from the fact that temperature scaling is not designed for these types of uncertainty, but specifically for the model's calibration. This behavior is demonstrated in the table below. While temperature scaling significantly improves the model's calibration on MNIST dataset as it is designed to do and as evidenced by the rAULC metric, it does not enhance the model's ability to detect OOD samples and may actually slightly deteriorate it. Also, as the reviewer noted, this method necessitates an additional hold-out set for post-hoc calibration, which might either require enlarging the dataset or could lead to deteriorated model performance. Moreover, the original temperature scaling is limited to classification tasks, with no clear method for extending it to regression tasks.
> > >
> > > |                     | Single | TempS | DeepE | ZigZag |
> > > |---------------------|--------|-------|-------|--------|
> > > | Accuracy ($\uparrow$) | 0.980  | 0.980 | 0.990 | 0.982  |
> > > | rAULC ($\uparrow$)    | 0.712  | 0.857 | 0.958 | 0.961  |
> > > | Size                 | 1x     | 1x    | 5x    | 1x     |
> > > | Inf. Time            | 1x     | 1x    | 5x    | 2x     |
> > > | Time                 | 1x     | 1x    | 5x    | 1x     |
> > > | ROC-AUC ($\uparrow$)  | 0.773  | 0.768 | 0.984 | 0.982  |
> > > | PR-AUC ($\uparrow$)   | 0.844  | 0.836 | 0.979 | 0.981  |
> > >
> > >
> > > * **"It would be nice to look at the code to see what's going on under the hood here."**
> > >
> > > We will add a link to a codes repository.
> > >
> > > * **"How is the "variance" / "standard deviation" estimated in regression?"**
> > >
> > > To depict uncertainty for Fig. 6 for a specific input $\mathbf{x}$, we computed $\mathbf{u} = |\mathcal{M}(\mathbf{x}, \mathbf{0}) - \mathcal{M}(\mathbf{x}, \mathcal{M}(\mathbf{x}, \mathbf{0}))|$ and then plotted the first model's prediction $\mathcal{M}(\mathbf{x}, \mathbf{0})$ as the final prediction, with $\pm \mathbf{u}$ indicating uncertainty through red shading. Similarly, for ensembles, uncertainty was represented by first computing predictions from several models, using the mean of these predictions as the final prediction, and depicting uncertainty with $\pm \sigma$ standard deviation through red shading. Thus, the $\mathbf{u} = |\mathcal{M}(\mathbf{x}, \mathbf{0}) - \mathcal{M}(\mathbf{x}, \mathcal{M}(\mathbf{x}, \mathbf{0}))|$ measure effectively represents the uncertainty (or standard deviation) used for our method in Fig.6 and other experiments.
> > >
> > > * **"I'm curious to know if 2-3 samples of ensembles are highly performant there as well."**
> > >
> > > We present uncertainty results for Imagenet for ensembles of 1, 2, and 3 models in the table below. As can be seen, under the same inference computational costs, our method outperforms ensembles in terms of uncertainty quality. Larger ensembles display improved metrics, but they necessitate significantly greater training and inference overheads.
> > >
> > > |                     | 1 Sample | 2 Samples | 3 Samples | ZigZag |
> > > |---------------------|----------|-----------|-----------|--------|
> > > | rAULC ($\uparrow$)  | 0.8      | 0.81      | 0.82      | 0.82   |
> > > | ROC-AUC ($\uparrow$) | 0.51     | 0.53      | 0.55      | 0.54   |
> > > | PR-AUC ($\uparrow$)  | 0.15     | 0.16      | 0.18      | 0.17   |
> > >
> > > * **"Presentation Comments."**
> > >
> > > We thank the reviewer for pointing out typos and suggesting ways to enhance the clarity of our paper. In the final version, we will incorporate the following changes: 1) adding labels to the axes in Figure 10 and making the points larger, 2) fixing the shading in tables and replacing it with bold and italic formatting, 3) correcting capitalization and reference issues in the related work and in Section 4.

---

> > > > ### Author Response · Authors · 2024-02-25
> > > > **Authors Response to Reviewer 9k95 (Cont'd)**
> > > >
> > > > References:
> > > >
> > > > [1] Barnard, E., Wessels, LFA. 'Extrapolation and Interpolation in Neural Network Classifiers.' IEEE Control Systems Magazine, 1992.
> > > >
> > > > [2] Haley, P. J., Soloway, D. 'Extrapolation Limitations of Multilayer Feedforward Neural Networks.' IJCNN International Joint Conference on Neural Networks, 1992.
> > > >
> > > > [3] Postels, J., et al. 'Deterministic Epistemic Uncertainty.', ICML, 2022.
> > > >
> > > > [4] Ovadia, Y., et al. 'Can You Trust Your Model's Uncertainty? Evaluating Predictive Uncertainty Under Dataset Shift.' NeurIPS, 2019.
> > > >
> > > > [5] Ashukha, A., et al. 'Pitfalls of in-domain uncertainty estimation and ensembling in deep learning', ICLR, 2020.
> > > >
> > > > [6] Durasov, N., et al. 'Masksembles for uncertainty estimation.', CVPR, 2021.
> > > >
> > > > [7] Wen, Y., et al. 'BatchEnsemble: An Alternative Approach to Efficient Ensemble and Lifelong Learning', ICLR, 2020.
> > > >
> > > > [8] Lakshminarayanan, B., et al. 'Simple and Scalable Predictive Uncertainty Estimation Using Deep Ensembles.', NeurIPS, 2017.
> > > >
> > > > [9] Malinin, A., et al. 'Predictive uncertainty estimation via prior networks.' , NeurIPS, 2018.
> > > >
> > > > [10] Maddox, W. J., et al. 'A simple baseline for bayesian uncertainty in deep learning.', NeurIPS, 2019.

---

> > > > ### Comment · Reviewer_9k95 · 2024-03-06
> > > > **Thanks for your responses**
> > > >
> > > > Thank you very much for your responses, they definitely resolved most of my concerns. Two points the authors should be aware of (I do not expect you to compare against these) for the future:
> > > >
> > > > - temperature scaling for regression is essentially https://proceedings.mlr.press/v80/kuleshov18a.html
> > > >
> > > > - "both models simultaneously with an uncertainty loss" I believe this is a typo - the idea in this type of approach is to first train the mean model, and then afterwards train the variance model. log \sigma^2 sometimes also has parameterization issues...

---

> > > > ### Comment · Reviewer_9k95 · 2024-03-06
> > > > **One small comment**
> > > >
> > > > I am a bit surprised that the direct approach of $u = | \mathcal{M}(x, 0) - \mathcal{M}(x, \mathcal{M}(x, 0)) |$ is roughly 2 standard deviations (or something approximating it) as I see in the paper's plots. Could the authors provide some intuition as to why this is the case?

---

> > > > > ### Author Response · Authors · 2024-03-06
> > > > > **Authors Response to Reviewer 9k95 (Cont'd)**
> > > > >
> > > > > We thank the reviewer for the valuable references and will definitely delve deeper into them in our future work.
> > > > >
> > > > > Regarding the variance connection, our intuition comes from the connection to auto-encoders mentioned in the related work. Following [Alain \& Bengio, JMLR 2014]'s formula $y_{1} - y_{0} = \mathcal{M}(x, \mathcal{M}(x, 0)) - \mathcal{M}(x, 0) \propto  \frac{\partial \log p(y_{0} | x)}{\partial y} + o(\sigma^{2}) $  where $p(y_{0}|x) $  is the model's implicit posterior distribution and $\sigma$ its standard deviation. Since $y_{0}$, as a MAP estimate, should be a mode of $p(y |x)$ and therefore close to a stationary point, the first term can be neglected, conceptually linking $|y_{1} - y_{0}|$ to prediction variance.

---

### Decision · Action_Editor_Sx9U · 2024-04-07

**Recommendation:** Accept with minor revision

**Comment:**

Reviewers found the paper to be generally clearly written, and to provide an interesting and effective alternative to techniques such as deep ensembles. While there were a number of requests for discussion and comparison to other techniques, these were adequately provided by the authors in the discussion period.

Following the author response, all three reviewers were in favour of acceptance. The AE upholds this consensus. We do however request that the authors incorporate in their manuscript some of the important points that arose in the discussion period.

**Requested changes**:
- add the discussion on the intuition behind the method, posted in the response to Reviewer t2aE
- include the new experimental results from the author response, including:
    - temperature scaling method
    - separate variance model
    - deep ensembles with 2x instead of 5x budget in Table 1
    - auto-encoder baseline
    - standard deviations of results
- (optional) consider moving Figure 1 to the top of page 2

**Audience:**

Estimating uncertainty from deep networks is a topic of considerable interest to the ML community; new techniques to do so cheaply are thus certainly of interest. Thus, we expect a good portion of the TMLR audience would find the results of interest.

**Claims And Evidence:**

The primary claim of the paper is that one can obtain reliable uncertainty estimates from deep networks, while requiring much less compute than existing approaches such as deep ensembles. This claim is demonstrated through several experiments on synthetic and real-world data, including image benchmarks and engineering applications such as predicting lift-to-drag ratios.

Reviewers raised questions about the conceptual underpinnings of the method, and the comparison to several natural baselines. Following a detailed author response, all three reviewers were sufficiently convinced of the method being sound and effective.

---

> ### Author Response · Authors · 2024-04-17
> **Thank you to the AE and Reviewers**
>
> We deeply appreciate the constructive feedback and insightful suggestions from the action editor and reviewers. We are immensely thankful to all reviewers for their invaluable contributions, which have notably strengthened our work. We just posted the camera-ready version of the paper, including these updates from previous discussions:
>
> * Added a discussion about the intuition behind the method at the beginning of the method section.
> * Included results on temperature scaling and a separate variance model in the ablations section (Tabs. 5 and 6).
> * Included an evaluation of Deep Ensembles with 2 samples instead of 5 in a new subsection of the ablations section (Tab. 4).
> * Added evaluations and discussions on auto-encoders in the supplementary material, as suggested by reviewer t2aE (Tab. 9).
> * Incorporated standard deviation results in the supplementary material (Tab. 7), referenced in the experiments section.
> * Added MNIST-S experiments requested by reviewer t2aE, detailed in the supplementary material (Tab. 8).
> * Moved Fig. 1 to the top of the second page.